# A redox mechanism underlying nucleolar stress sensing by nucleophosmin

Kai Yang[1], Ming Wang[1], Yuzheng Zhao[2], Xuxu Sun[1], Yi Yang[2], Xie Li[2], Aiwu Zhou[3], Huilin Chu[1], Hu Zhou[4], Jianrong Xu[5], Mian Wu[6], Jie Yang[1] & Jing Yi[1]

The nucleolus has been recently described as a stress sensor. The nucleoplasmic translocation of nucleolar protein nucleophosmin (NPM1) is a hallmark of nucleolar stress; however, the causes of this translocation and its connection to p53 activation are unclear. Using single live-cell imaging and the redox biosensors, we demonstrate that nucleolar oxidation is a general response to various cellular stresses. During nucleolar oxidation, NPM1 undergoes S-glutathionylation on cysteine 275, which triggers the dissociation of NPM1 from nucleolar nucleic acids. The C275S mutant NPM1, unable to be glutathionylated, remains in the nucleolus under nucleolar stress. Compared with wild-type NPM1 that can disrupt the p53–HDM2 interaction, the C275S mutant greatly compromises the activation of p53, highlighting that nucleoplasmic translocation of NPM1 is a prerequisite for stress-induced activation of p53. This study elucidates a redox mechanism for the nucleolar stress sensing and may help the development of therapeutic strategies.

[1] Shanghai Key Laboratory for Tumor Microenvironment and Inflammation, Key Laboratory of Cell Differentiation and Apoptosis of Chinese Ministry of Education, Department of Biochemistry and Molecular Cell Biology, Shanghai Jiao Tong University School of Medicine, 280 South Chongqing Road, Shanghai 200025, China. [2] Synthetic Biology and Biotechnology Laboratory, State Key Laboratory of Bioreactor Engineering, School of Pharmacy, East China University of Science and Technology, 130 Mei Long Road, Shanghai 200237, China. [3] Department of Pathophysiology, Shanghai Jiao Tong University School of Medicine, 280 South Chongqing Road, Shanghai 200025, China. [4] Shanghai Institute of Materia Medica, 555 Zu Chong Zhi Road, Zhang Jiang Hi-Tech Park, Shanghai 201203, China. [5] Department of Pharmacology, Shanghai Jiao Tong University School of Medicine, 280 South Chongqing Road, Shanghai 200025, China. [6] School of Life Sciences, University of Science and Technology of China, 96 Jinzhai Road, Hefei, Anhui 230022, China. Correspondence and requests for materials should be addressed to J. Yang (email: yangjieyj@shsmu.edu.cn) or to J. Yi (email: yijing@shsmu.edu.cn).

The nucleolus is a nuclear compartment for ribosome biogenesis where the transcription and processing of pre-ribosomal RNA and the assembly of ribosomal subunits take place. However, studies over the past two decades have revealed non-ribosomal functions for the nucleolus[1–4]. Proteomic analysis has identified a large number of nucleolar proteins that are involved in diverse cellular processes, including ribosomal biogenesis, cell cycle control and stress signalling[5–7]. Remarkably, many roles for the nucleolus are regulated by the sequestration or release of specific nucleolar proteins[8] and various cellular stresses are often accompanied by dramatic changes in the structural organization and protein composition of the nucleoli[7]. However, it remains unknown how the various stress inducers lead to the release of nucleolar proteins into the nucleoplasm or the cytoplasm, a process that is often called 'relocalization' or 'translocation.' Thus, although recognizing the nucleolus as a stress sensor is a dramatic revision of our conception of the nucleolus, how the nucleolus senses stress has yet to be understood.

Nucleophosmin (NPM1) or B23 is an abundant nucleolar protein that can frequently shuttle between the nucleolus and the nucleoplasm or cytoplasm, interacting with a plethora of macromolecules to play a multifunctional role in cells[9]. Notably, NPM1 translocation from the nucleolus to the nucleoplasm is the most typical hallmark of nucleolar stress[10]. Nucleolar stress originally referred to the stressful events that impair the homeostasis of ribosomal biogenesis and activate the cellular stress response, which is also called ribosomal stress[11,12], and it is currently defined as alterations in the nucleolar structure and function that are induced by a variety of abnormal conditions[13]. NPM1 translocation appears to be a general event in nucleolar stress responses, even for stresses induced by different factors[14–16]. Moreover, NPM1 might be an upstream signalling molecule that links various stresses with p53 accumulation[12,16,17]. However, the mechanism underlying this well-known phenomenon of stress-induced NPM1 translocation remains unclear.

NPM1 has been identified as a phosphoprotein, as it is phosphorylated during specific cellular events[18]. The distinct phosphorylation patterns of NPM1 could be correlated with the regulation of NPM1 localization[19]. Many other types of posttranslational modifications (PTMs) of NPM1 have also been reported[20–22]. These PTMs might presumably act as regulatory mechanisms underlying NPM1 translocation; however, none of the reported PTMs has been linked to the regulation of NPM1 translocation in response to nucleolar stress.

Using a combination of a number of innovative technologies, including live-cell imaging and a genetically encoded redox biosensor, we provide evidence addressing these questions and uncover a sensing mechanism for NPM1 in response to nucleolar and other stresses. We demonstrate that nucleolar oxidation is a general response to various cellular stresses. Accompanying the redox changes in the nucleolus, NPM1 undergoes S-glutathionylation on cysteine 275, which triggers the nucleoplasmic translocation of NPM1. This translocation can be attributed to the dissociation of NPM1 from nucleolar nucleic acids following S-glutathionylation, which is demonstrated by in vitro and in vivo NPM1–RNA/DNA interaction assays. The C275S mutant, which is unable to be glutathionylated and thus remained in the nucleolus, greatly compromises p53 stabilization under typical nucleolar stress conditions induced by actinomycin, regardless of the presence of the ribosomal protein L23 or ARF. Only wild-type (WT) NPM1 or a mutant localized to the nucleoplasm is able to lead to p53 accumulation under nucleolar stress, highlighting the fact that the nucleoplasmic translocation of NPM1 is a prerequisite for stress-induced p53 activation.

## Results

**Nucleolar oxidation is a general cellular stress response.** Intracellular reactive oxygen species (ROS) can be increased by many stimuli[23,24]. As NPM1 translocation has been observed under various cellular stress conditions in addition to typical nucleolar stress[14,16], we hypothesized that these stress inducers might produce a redox change in the nucleolar compartments. We constructed a nucleus-specific ratiometric redox probe based on reduction–oxidation-sensitive green fluorescent protein (roGFP1)[25]. This nucleus-specific roGFP1 (NLS-roGFP1) fluoresced throughout the entire nucleus (Fig. 1a) and its distribution remained unchanged under oxidative stress conditions (Supplementary Fig. 1a). The cells expressing NLS-roGFP1 were then challenged by a series of stressors including hydrogen peroxide ($H_2O_2$), hypoxia, ultraviolet irradiation, heat shock, starvation (Earle's balanced salt solution culture) and actinomycin D (Act.D). The nucleoli of all of the cells underwent a rapid oxidation to varying extents; however, these redox disturbances could be partially prevented by pretreatment with the anti-oxidant N-acetyl-cysteine (NAC) (Fig. 1b). These data indicated that an increase in ROS generation in the nucleolar compartment occurs universally under various cellular stresses. Given that the individual cells showed marked differences in the extent of the redox change, single live-cell imaging was used to monitor the real-time process of nucleolar oxidation. Two stress conditions were selected: Act.D (an inducer of nucleolar stress via the inhibition of RNA polymerase I[11]) and $H_2O_2$ (a membrane-permeable ROS that is a general inducer of cellular oxidative stress). Nucleolar oxidation began 30 min after Act.D treatment and was aggravated at 1 h (Fig. 1c). Strikingly, $H_2O_2$ elicited the oxidation of the nucleolus within the first minute. This oxidation progressed and was maintained for at least 30 min but was attenuated by NAC pretreatment (Fig. 1d,e). Thus, nucleolar oxidation is indeed a general early event in response to various stresses, including typical nucleolar stress. In addition, we examined more cell types following their exposure to 500 μM $H_2O_2$ treatment, including the non-tumour cell line HEK293, the osteosarcoma cell line U2OS and the gastric carcinoma cell line MKN45. The oxidation of the nucleolus was recorded in live cells within 10 min (Supplementary Fig. 1b), suggesting that this is a general phenomenon.

**NPM1 translocation is elicited by nucleolar oxidation.** We used immunofluorescence to observe endogenous NPM1 translocation in cells challenged by the aforementioned stress conditions. The relative fluorescence intensity (RFI; see 'Methods') in nucleoli showed that NPM1 was largely dispersed to the nucleoplasm under all of the stress conditions tested. The antioxidant NAC was able to attenuate or even completely prevent this translocation (Fig. 2a and Supplementary Fig. 2a), suggesting that the nucleoplasmic translocation of NPM1 was induced by nucleolar redox changes. We next visualized the dynamics of EGFP-NPM1 translocation in response to $H_2O_2$ treatment in single live cells. The histone subunit H2B was labelled with mCherry, to show the shape of the nucleus and to control the focal plane in that cell. Treatment with 500 μM $H_2O_2$ produced a dispersion of nucleolar EGFP-NPM1 to the nucleoplasm that lasted for at least 30 min, resulting in a decrease in nucleolar NPM1 (Fig. 2b). Treatments with increasing doses of $H_2O_2$ (250 and 500 μM) caused a sustained loss of nucleolar NPM1 intensity in a dose-dependent manner. The RFI following 500 μM $H_2O_2$ treatment was decreased to 60%; in contrast, a lower concentration of $H_2O_2$ (125 μM) produced only a slight, transient loss within the first few minutes (Fig. 2c). Pretreatment with NAC restored the nucleolar

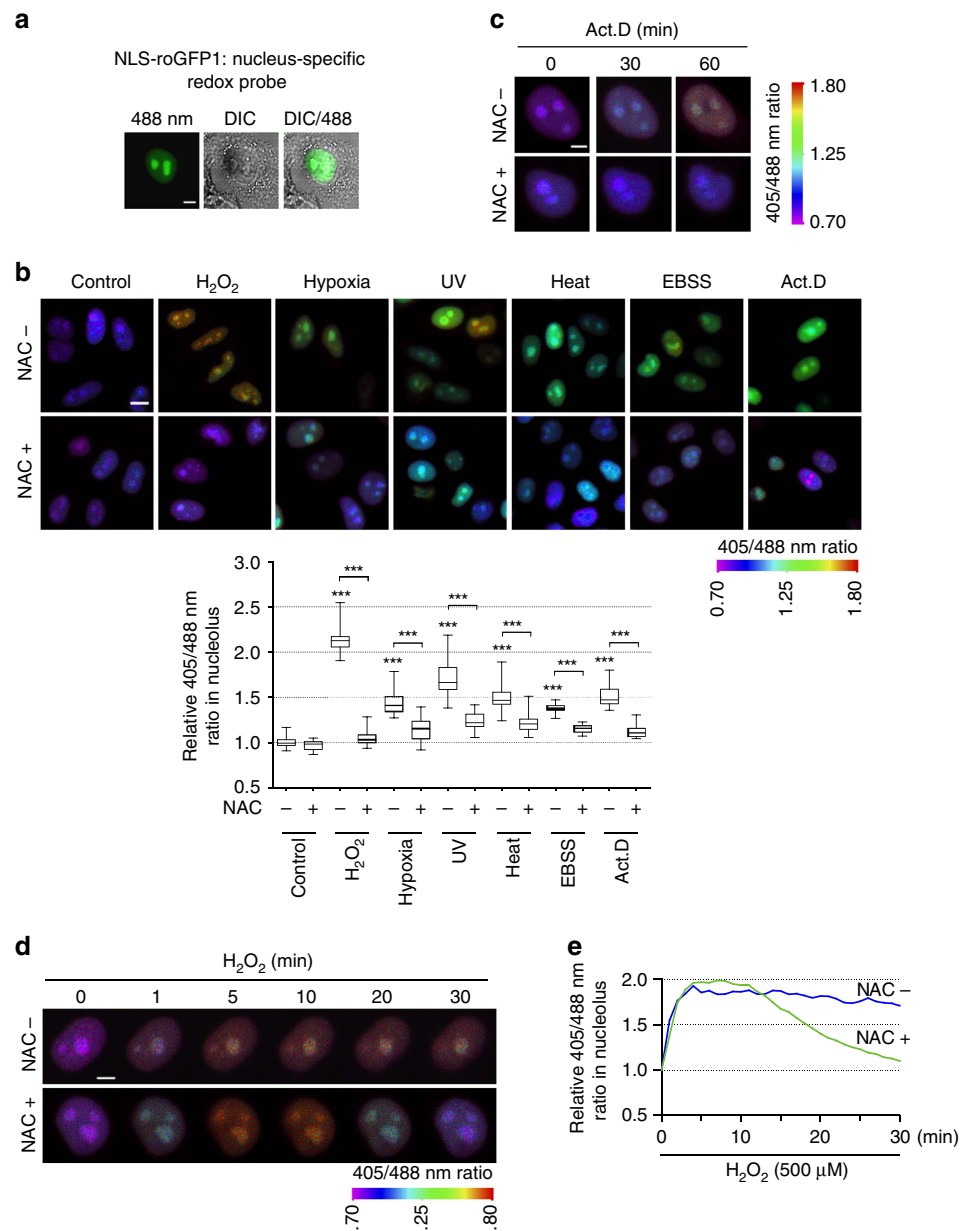

**Figure 1 | Nucleolar oxidation is a general cellular stress response.** (**a**) Nuclear localization of redox indicator NLS-roGFP1 in HeLa cells, indicated with DIC image. (**b**) Nucleolar redox changes indicated by NLS-roGFP1 in HeLa cells exposed to $H_2O_2$ (500 μM, 30 min), hypoxia (1% $O_2$, 1 h), ultraviolet irradiation (100 J m$^{-2}$), heat shock (42 °C, 30 min), Earle's balanced salt solution (EBSS; 6 h) or Act.D (8 nM, 1 h) with or without (±) 4 h of the antioxidant NAC (5 mM) pretreatment. Representative pseudocolourful images (upper, see Image processing and fluorescence signal quantifications for details) were shown and relative nucleolar ratio values were displayed (bottom). $n = 30$ cells. Unpaired $t$-test, ***$P < 0.001$ with respect to treated versus untreated or −NAC versus +NAC cells and $P < 0.001$ with any of the NAC-treated cells versus control NAC-treated cells. (**c,d**) Nucleolar redox changes in HeLa cell exposed to Act.D (8 nM, **c**) or $H_2O_2$ (500 μM, **d**) ±NAC pretreatment, monitored by live-cell imaging. (**e**) Relative nucleolar ratio values changes per minute in **d**. Data were represented as mean ± s.e.m. Scale bars, 5 μm (**a,c,d**) and 10 μm (**b**).

intensity (Fig. 2d), whereas the protein disulfide reductant dithiothreitol (DTT) produced a more powerful effect in blocking NPM1 translocation compared with NAC; a complete nucleolar restoration of NPM1 was achieved by DTT (Fig. 2e). To ensure that the GFP tag did not significantly impair translocation efficiency, we examined the localization of a FLAG-tagged NPM1 by immunofluorescence. The RFI of nucleolar FLAG-tagged NPM1 was decreased following 500 μM $H_2O_2$ treatment slightly more effectively (decreased to 40%; Supplementary Fig. 2b). Nevertheless, both tagged proteins produced similar results as the endogenous setting shown in Fig. 2a. The nucleoplasmic

translocation of NPM1 was also examined in additional types of live cells; decreases in nucleolar RFI, which ranged from ∼25 to 55%, were observed within 20 min (Supplementary Fig. 2c), suggesting that this is a general phenomenon.

To determine the causal relationship between nucleolar redox change and NPM1 translocation, the NLS-roGFP1 probe and mCherry-NPM1 were co-transfected into the same cell. The addition of $H_2O_2$ caused an immediate and stepwise nucleolar oxidation that was accompanied by NPM1 translocation. However, when DTT was co-administered, no translocation of NPM1 occurred, although the redox probe

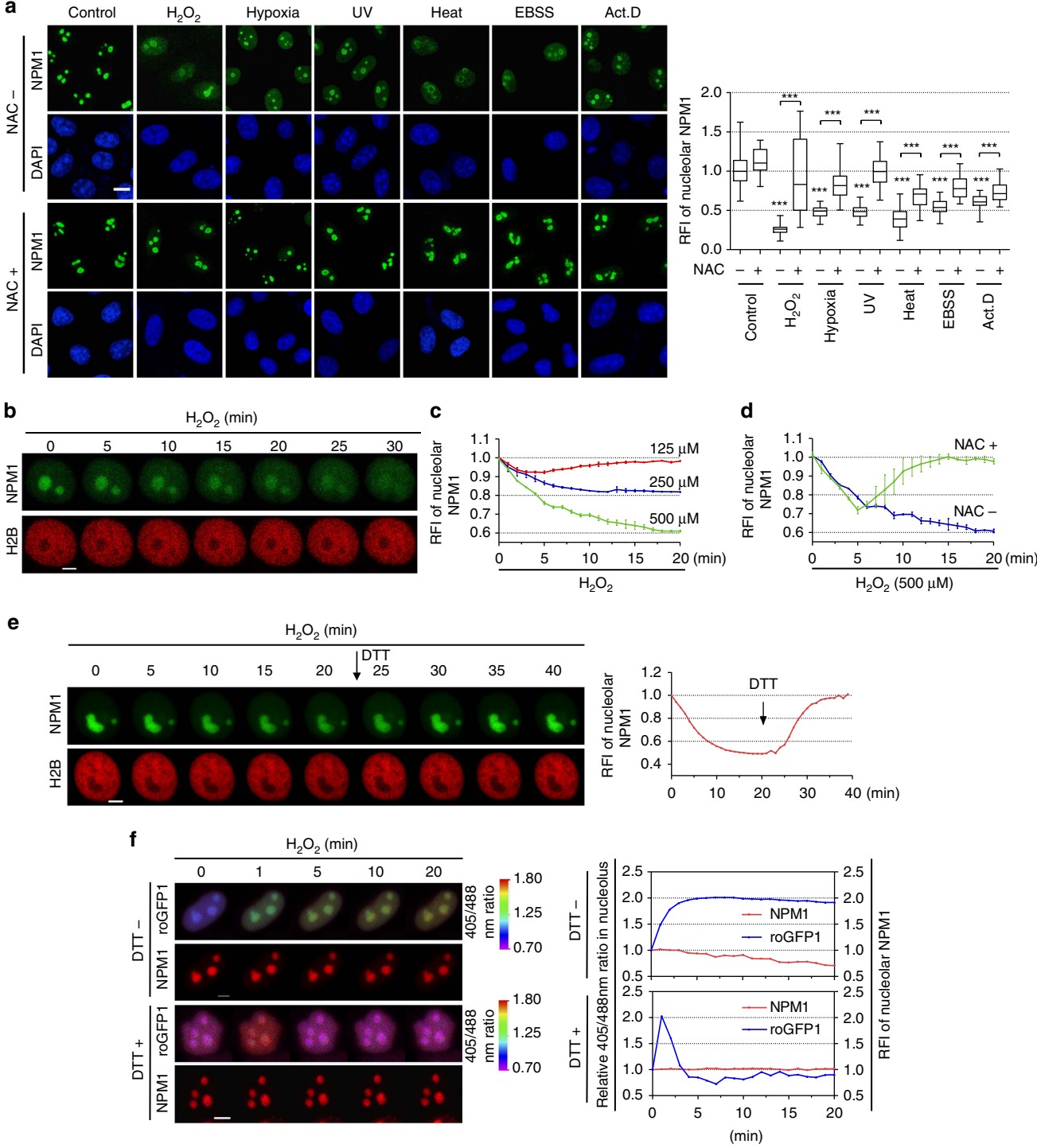

**Figure 2 | NPM1 translocation is elicited by nucleolar oxidation.** (**a**) Endogenous NPM1 translocation under various cellular stresses ± NAC pretreatment (as in Fig. 1b). Representative images (left) and the RFI of the nucleolar NPM1 (right) were displayed. $n = 55$ cells. Unpaired $t$-test, ***$P < 0.001$ with respect to treated versus untreated or $-$NAC versus $+$NAC cells. The nucleolar and nucleoplasmic line profiles of NPM1 were analysed in Supplementary Fig. 2a. (**b**) EGFP-NPM1 translocation upon $H_2O_2$ (500 μM) exposure, visualized by live-cell imaging. Representative images at indicated times were displayed. (**c,d**) RFI changes of nucleolar EGFP-NPM1 per minute upon three doses of $H_2O_2$ treatments or $H_2O_2$ (500 μM) plus pretreatment of NAC. (**e**) EGFP-NPM1 translocation upon $H_2O_2$ (500 μM) exposure followed by the addition of 5 mM DTT after 20 min. (**f**) EGFP-NPM1 translocation and nucleolar redox changes upon $H_2O_2$ (500 μM) exposure ± DTT pretreatment in the same cells. Concurrent changes of relative 405/488 nm ratio values reflecting redox state (left $y$ axis) and RFI of nucleolar EGFP-NPM1 (right $y$ axis) per minute were plotted. Scale bars, 10 μm (**a**) and 5 μm (**b,e,f**). Data were represented as mean ± s.e.m.

indicated a slight transient oxidation of the nucleolus (Fig. 2f). These data confirmed that the shuttling of NPM1 between the nucleolus and nucleoplasm might be controlled by the redox states of the two compartments, and that its nucleoplasmic translocation following stress is elicited by nucleolar oxidation.

**Cys[275] oxidation is responsible for NPM1 translocation.** The DTT reversal experiments strongly implied a correlation between the reversible oxidative modification of proteins and stress-induced NPM1 translocation. To validate the association of cysteine thiol modification with translocation and to map the potential modification site(s), we constructed a 3C/S mutant EGFP-NPM1 that eliminated thiol modification sites by substituting all three cysteine residues (Cys[21], Cys[104] and Cys[275]) with serines. A WT mCherry-NPM1 was co-transfected into the same cells to compare the translocation dynamics. Unlike the WT protein, mutant 3C/S was resistant to translocation (Fig. 3a). Cysteine mutagenesis was then performed in pairs and only the mutants containing C275S failed to respond to H$_2$O$_2$-induced translocation (Fig. 3b), revealing that Cys[21] or Cys[104] was not

responsible for this process. Next, a C275S single mutant was examined, which indeed lost the ability to translocate following stress in a way that was similar to the 3C/S mutant (Fig. 3c). This finding was also verified in the EGFP-C275S NPM1 and mCherry-H2B co-transfected living cell system (Supplementary Fig. 3a), and in a FLAG-C275S NPM1 immunofluorescence assay (Supplementary Fig. 3b). These results support the hypothesis that oxidative modification on Cys[275] is essential for redox-regulated NPM1 nucleoplasmic translocation.

Cys[275] is within the C-terminal domain that is important for nucleolar localization[26]. The mutation of two tryptophans in this domain, Trp[288] and Trp[290], into alanines led to nucleoplasmic localization[26–28]. We therefore speculated that the oxidative modification on Cys[275] altered NPM1 localization by interfering with the nucleolar localization function of the carboxy-terminal domain. If this were the case, the explanation for the nucleolar retention of C275S under stress might be the absence of the thiol modification and the consequent interference, rather than the acquisition of nucleolar retention. Therefore, we predicted that the additional mutagenesis of two tryptophans (leading to the loss of nucleolar localization) would abolish the nucleolar retention of C275S. To test this, we constructed a W288/290A-C275S mutant that was tagged with FLAG and the localization of this NPM1 triple mutant was examined by immunofluorescence using an anti-FLAG antibody. We found that the W288/290A-C275S triple mutant remained localized to the nucleoplasm, behaving similarly to the W288/290A double mutant (Fig. 3d). These data strongly indicated that the stress-induced NPM1 translocation might be a consequence of the disruption of the C-terminal function of nucleolar localization, similar to what was caused by the W288/290A mutant.

**NPM1 is S-glutathionylated at Cys[275] under stress.** Reversible thiol oxidative modifications to cysteine include the formation of disulfide, sulfenic and sulfinic acids, and S-glutathionylation, which can change protein conformation or alter interactions with other macromolecules. Therefore, we investigated the type of oxidative modification occurring on Cys[275]. As the responsible cysteine residue was single, the possibility of a formation of an intramolecular disulfide bond was excluded. The oligomerization (to form pentamers) of NPM1 was unaffected by redox changes[29]; therefore, the formation of intermolecular disulfides between two or more NPM1 was also unlikely. To mimic the negative charge of sulfenic acid, Cys[275] was replaced with aspartic acid. This C275D mutant remained localized in the nucleolus under non-stress conditions and was resistant to being translocated following stress (Supplementary Fig. 4a). Thus, modification with sulfenic acid on Cys[275] was not possible either. Although a disulfide based on Cys[275] of NPM1 and another cysteine of other proteins could not be excluded, we first turned to consider S-glutathionylation on this residue.

S-glutathionylation is a reversible but relatively stable oxidative modification containing a disulfide bond between the cysteine thiol of a protein and glutathione (GSH). We performed a co-immunoprecipitation (co-IP) assay and detected a band slightly larger than 38 kDa (NPM1) in the precipitates using antibodies against either GSH or NPM1. The expression of these bands was enhanced by H$_2$O$_2$ in a dose-dependent manner and NAC abolished the enhancement of this modification (Fig. 4a). These modification bands were similarly detectable in other cell types exposed to the treatments (Supplementary Fig. 4b). These data indicate that NPM1 is glutathionylated under conditions of oxidative stress. Next, we used mass spectrometry to analyse the samples exposed to H$_2$O$_2$, verifying the existence of

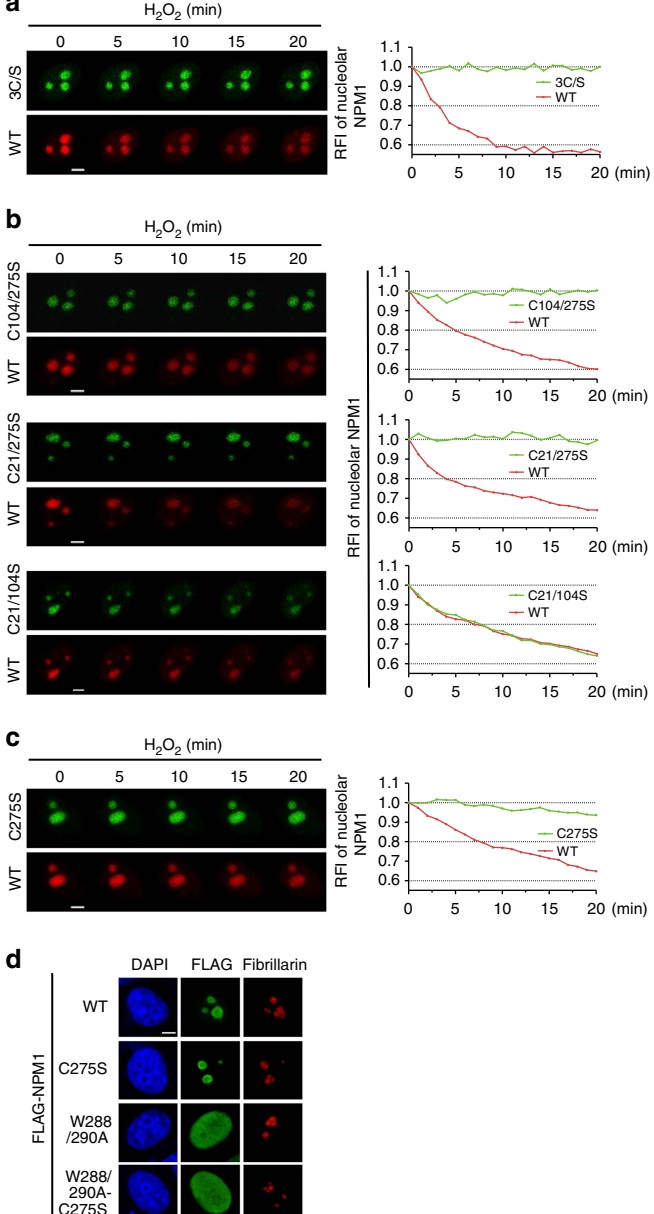

**Figure 3 | Cys[275] oxidation is responsible for NPM1 translocation.** (**a–c**) Translocation of mCherry-NPM1 WT and EGFP-NPM1 mutants in identical cells upon H$_2$O$_2$ (500 μM) exposure, visualized by live-cell imaging for 20 min. (**d**) Localization of various FLAG-NPM1 mutants in unstressed HeLa cells. Scale bars, 5 μm. 3C/S, C21/104/275S.

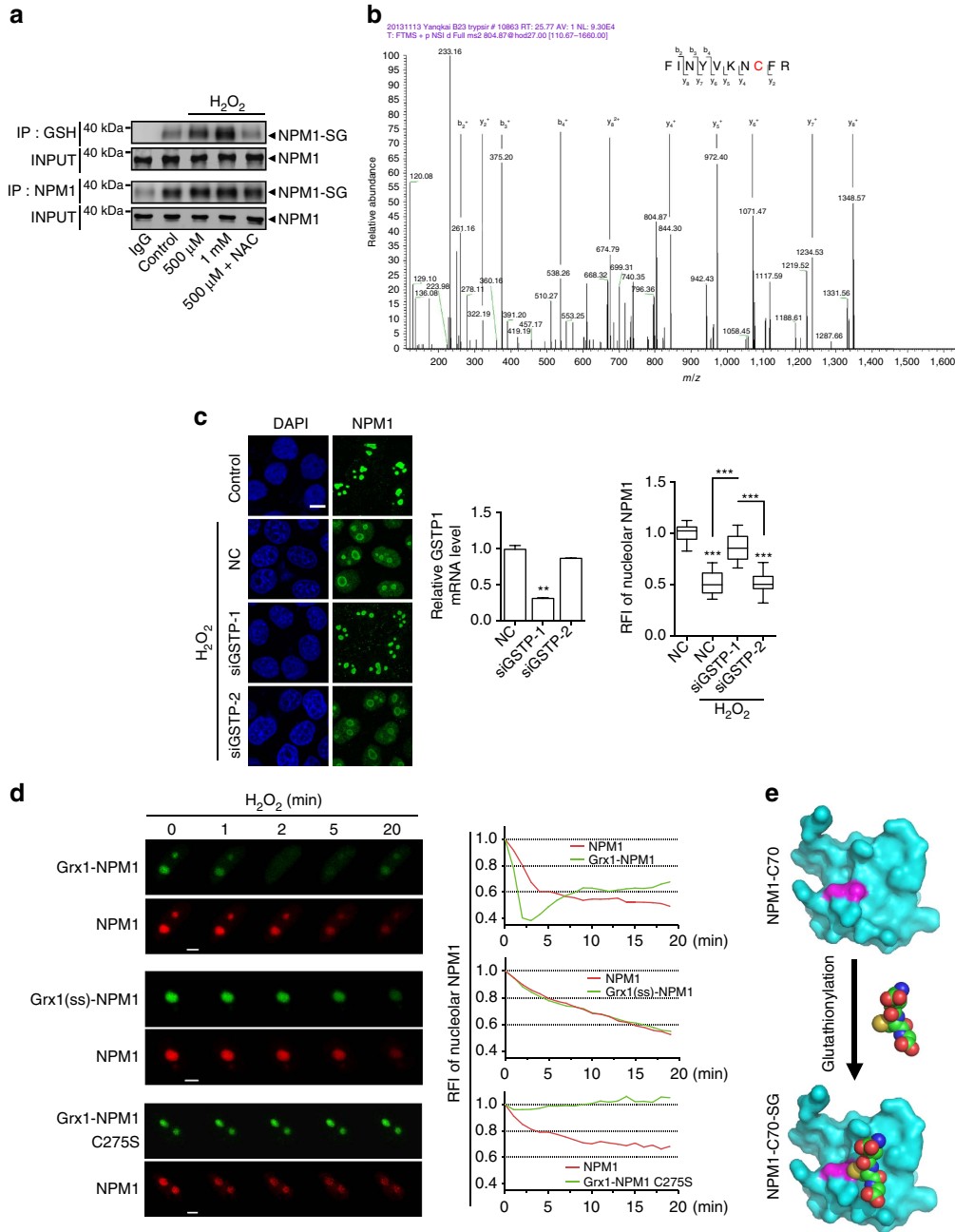

**Figure 4 | NPM1 is *S*-glutathionylated at Cys[275] under stress. (a)** Non-reduced co-IP assay. After gradient doses of $H_2O_2$ (500 μM, 1 mM) treatments ± antioxidant NAC (5 mM) pretreatment, *S*-glutathionylated NPM1 in HeLa cells was co-immunoprecipitated and analysed with either anti-GSH or anti-NPM1 antibodies. **(b)** Tandem mass spectrometry (MS/MS) analysis of NPM1 protein derived from non-reduced co-IP. FLAG-NPM1 was transfected to HEK293T cells exposed to $H_2O_2$ (500 μM) and immunoprecipitated by anti-FLAG M2 gel. The amino acid sequence of peptides was reconstructed by analysis of the $b^-$ ion or $y^-$ ion series. The MS/MS spectrum of the $m/z$ $y_5^+$ (972.4026) was corresponding to the NPM1 peptide KNCFR *S*-glutathionylated at Cys[275], in comparison with the non-modified spectrum 667.3344 in Supplementary Fig. 4c. **(c)** NPM1 translocation (left) and RFI changes of nucleolar NPM1 (right, $n = 57$ cells) after $H_2O_2$ (500 μM) treatment in HeLa cells. GSTP were silenced with two siRNA oligo and the relative GSTP mRNA level were showed (middle, siGSTP-1 and siGSTP-2). Unpaired *t*-test, **$P < 0.01$ or ***$P < 0.001$. **(d)** Translocation of EGFP-NPM1 WT and C275S mutant fused with active or inactive Grx1 at N-terminal upon $H_2O_2$ (500 μM) exposure. **(e)** The surface representation of NPM1 binding interface was based on NPM1-C70 structure (PDB *2vxd*) with Cys[275] coloured in magenta. GSH molecule, shown in sphere with nitrogen atom in blue, oxygen in red and carbon atom in green, could covalently link to Cys[275] through thiol exchange. Scale bars, 10 μm **(c)** and 5 μm **(d)**. Data were represented as mean ± s.e.m. Grx1(ss), inactive Grx1. NPM1-SG, *S*-glutathionylated NPM1. NPM1-C70, C-terminal of NPM1 (70 amino acids).

the *S*-glutathionylation of NPM1 at Cys[275] (Fig. 4b and Supplementary Fig. 4c).

Glutathione *S*-transferase Pi (GSTP) catalyses the forward reaction of *S*-glutathionylation[30]. We knocked down GSTP expression using small interfering RNA (siRNA) with varying efficiencies. Based on the level of remaining GSTP messenger RNA, the $H_2O_2$-induced NPM1 translocation to the nucleoplasm could be completely or minimally blocked (Fig. 4c). These data verified that *S*-glutathionylation is a cause of NPM1 translocation. Glutaredoxin 1 (Grx1) is a thiol oxidoreductase that selectively

interacts with GSH[31]. In previous reports, a Grx1-roGFP2 fusion protein was constructed to monitor the dynamics of reduced GSH versus oxidized GSH (GSSG)[32–34]. We therefore constructed an EGFP-Grx1-NPM1 fusion protein and observed its translocation dynamics in a live-cell imaging system. Compared with the control mCherry-NPM1 that was co-expressed in the same cells, nucleolar EGFP-Grx1-NPM1 showed an earlier and more thorough dispersion following $H_2O_2$ treatment with a short recovery period (Fig. 4d, top). In contrast, catalytically inactive Grx1-fused NPM1 (ref. 34), Grx1(ss)-NPM1, showed translocation dynamics that were indistinguishable from those of the control mCherry-NPM1 (Fig. 4d, middle), confirming that the drastic and quick translocation of Grx1-NPM1 was indeed caused by the Grx1-catalysed NPM1 S-glutathionylation cycle. The Grx1-NPM1 C275S mutant further verified that Grx1-catalysed NPM1 S-glutathionylation occurred on Cys[275] (Fig. 4d, bottom). Using Grx1-roGFP2 as a biosensor for the GSH/GSSG ratio[34], we found a drastic increase in the levels of GSSG following $H_2O_2$ treatment (Supplementary Fig. 4d). These results, together with the preceding data, strongly suggest that S-glutathionylation on Cys[275] impedes NPM1 nucleolar localization under conditions of oxidative stress.

The C-terminal domain of NPM1, which encompasses the last 70 residues of NPM1 (NPM1-C70), residues 225–294, is critical for its interaction with nucleolar nucleic acids[35]. The solution structure of NPM1-C70 has been previously solved, showing a surface through which NPM1 might associate with nucleolar binding partners[26]. Interestingly, Cys[275] is located near the centre of this interface[26]. Based on this structure, we simulated the covalent modification of Cys[275] by a bulky GSH molecule. The simulated docking indicated that the GSH molecule, apart from forming a disulfide linkage with Cys[275], could also form stabilizing interactions with residues around Cys[275]. This GSH modification might impede the association of the interface with the potential nucleolar macromolecules (Fig. 4e), which could explain the translocation of NPM1 following stress.

**S-glutathionylated NPM1 dissociates from rRNA/rDNA.** How the C-terminal domain determines the nucleolar localization of NPM1 remains unclear, but binding with rRNA is widely believed to be critical[36,37]. Two studies have reported that NPM1 could also bind to nucleolar rDNA chromatin[38,39]. We hypothesized that the nucleoplasmic translocation of NPM1 under stress might be due to its dissociation from these nucleic acids. The cells were incubated with RNase or Dnase, to digest rRNA or rDNA, respectively. Both of the treatments caused a dramatic dispersion of NPM1 into the entire nucleoplasm (Fig. 5a, left). These data suggested that the normal NPM1 nucleolar localization relies on the direct or indirect anchoring of NPM1 to both rRNA and rDNA, although it remains possible that RNase treatment could cause the destruction of pre-ribosomal particles, which could also account for the diffusion of NPM1 into the nucleoplasm. Digestion experiments were then performed using the C275S mutant and $H_2O_2$ treatments. Following stress, the C275S mutant remained in the nucleoli but dispersed into the nucleoplasm after RNase or DNase digestions, indicating that the mutant protein moved out of the nucleoli whenever the nucleic acids disappeared (Fig. 5a, right). We used RNA immunoprecipitation (RIP) and chromatin immunoprecipitation (ChIP) assays to quantitatively compare the ability of NPM1 to bind with rRNA and rDNA before and after $H_2O_2$ treatment. Primers were designed for three rRNA sequences targeting 47S, 18S and 28S transcripts, and for six rDNA sequences covering the rRNA gene promoter and coding region (Supplementary Fig. 5a). Equal quantities of FLAG-NPM1 were efficiently precipitated under the various conditions

(Fig. 5b–e) and the quantities of rRNA and rDNA in the precipitates were assessed by real-time reverse transcriptase–PCR and PCR, respectively, along with several internal controls (Supplementary Fig. 5b–d). All three rRNA products bound to NPM1, especially products I and III, were markedly decreased under stress conditions compared with controls and these decreases were largely prevented by NAC (Fig. 5b). Likewise, all six rDNA products bound with NPM1 were decreased under stress and these decreases were also reversed by NAC (Fig. 5c). These results demonstrated that NPM1 becomes dissociated from rRNA and rDNA under conditions of stress. Unlike WT NPM1, the binding capacity to either rRNA or rDNA of the C275S mutant was not decreased under oxidative stress (Fig. 5d,e). These data indicate that the nucleolar retention of C275S under stress is due to the fact that the mutant protein has lost its ability to undergo thiol modification, thereby becoming unable to dissociate from the nucleic acids.

RIP assays were also performed to assess whether GSSG could promote the dissociation of NPM1 from nucleic acids. The amount of rRNA bound to NPM1 was less in the GSSG-containing sample than in the control (Supplementary Fig. 5e).

Next, we tested whether S-glutathionylation could block the interactions between NPM1, and both rDNA and rRNA in vitro. The amino and C termini of NPM1 truncates N224 (1–224) and C70 (225–294) were constructed and their subcellular localization was compared. Immunostaining revealed that N224 showed no nucleolar localization, whereas C70 was normally distributed in the nucleoli, nucleoplasm and cytoplasm (Supplementary Fig. 5f). This finding verified that NPM1-C70, which, unlike NPM1-N224, mimics the nucleolar localization of full-length NPM1, should be used to observe the in vitro interaction with nucleolar nucleic acids. NPM1-C70 and the mutant C275S were expressed in prokaryotic cells and purified. Previously, Chiarella et al.[35] found a significant number of putative G-quadruplex-forming sequences on the non-template of the rDNA gene that encodes the 47S rRNA precursor and verified that the G-quadruplexes formed by these sequences had a greater affinity for NPM1-C70. We therefore selected one of the putative G-quadruplex-forming sequences in the 47S rRNA precursor to represent the rDNA nucleic acid and its rRNA counterpart to represent the rRNA nucleic acid. We designed the surface plasmon resonance (SPR) assay similar to the experiments previously conducted by Chiarelle et al.[35] and tested whether the interaction of NPM1-C70 and these G-quadruplexes could be interrupted following the addition of GSSG in vitro. Our results showed that GSSG blocked the interaction of NPM1-C70 with rDNA G-quadruplex (Fig. 6, G4-DNA, WT versus WT + GSSG; Supplementary Fig. 5g and Table 1 for the $K_D$ values). Importantly, the interaction of the translocation-disabled C275S mutant with the rDNA G-quadruplex was unaffected by GSSG (Fig. 6, G4-DNA, C275S versus C275S + GSSG; Supplementary Fig. 5g and Table 1). The measurement of the interactions of NPM1-C70 WT or C275S with rRNA G-quadruplex indicated similar effects (Fig. 6, G4-RNA; Supplementary Fig. 5g and Table 1).

Supporting the model proposed in Fig. 4e, these data provide evidence that the S-glutathionylation on Cys[275] forms a steric obstacle between NPM1 molecules and nucleolar DNA and/or RNA. This obstacle may be sufficient to disassociate NPM1 from nucleolar nucleic acids and thereby trigger its nucleoplasmic translocation.

**NPM1 translocation is required for p53 activation.** The NPM1 translocation and activation of the master stress-response protein p53 have often been described concomitantly under nucleolar and other stresses[12,16,17]. We next addressed the question of whether and how p53 stabilization is dependent on the translocation of

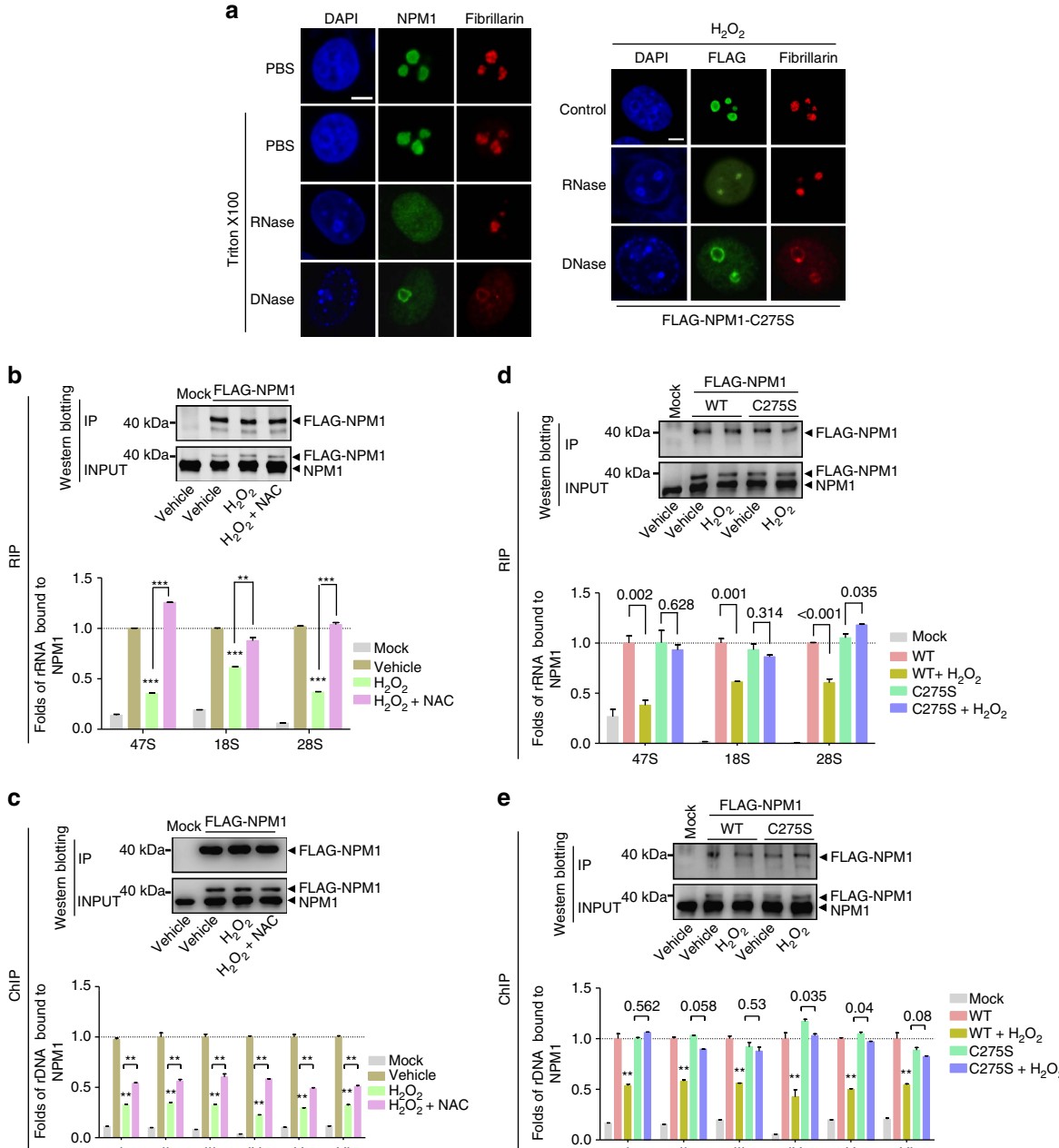

**Figure 5 | NPM1 dissociates from rRNA/rDNA following S-glutathionylation.** (**a**) Nucleoplasmic dispersion of endogenous NPM1 (left) or FLAG-NPM1 C275S (right) after RNase A (1 mg ml$^{-1}$) and DNase I (0.5 U ml$^{-1}$) digestions in HeLa cells with or without H$_2$O$_2$ (500 μM) treatment. (**b,c**) Equal quantities of FLAG-NPM1 immunoprecipitated by anti-FLAG M2 gel in RIP (**b**) and ChIP (**c**) assays in HEK293T cells treated with H$_2$O$_2$ (500 μM) ± NAC pretreatment, blotted by anti-NPM1 antibody (upper). The relative quantities of rRNA and rDNA bound with these FLAG-NPM1 were assessed (bottom). Unpaired t-test, \*\*$P < 0.01$ or \*\*\*$P < 0.001$ with respect to treated versus untreated or −NAC versus +NAC cells. (**d,e**) Similar experiments to **b,c**; the relative quantities of rRNA (**d**) and rDNA (**e**) bound with FLAG-NPM1 WT or mutant C275S in HEK293T cells exposed to H$_2$O$_2$ (500 μM) were assessed. \*\*$P < 0.01$ or showed above the bars. In **b–e**, Mock referred to the transfection negative control, whereas Vehicle referred to the treatment negative control. Data were represented as mean ± s.e.m. Scale bars, 5 μm. The internal controls of RIP and ChIP assays are shown in Supplementary Fig. 5b–d.

NPM1. First, we determined the endogenous levels and localization of NPM1 and p53 using immunofluorescence in U2OS cells challenged with the most commonly used inducer for nucleolar stress, Act.D. As expected, Act.D elicited concurrent NPM1 translocation and p53 accumulation in the nucleoplasm, which could be attenuated by NAC or completely reversed by DTT (Fig. 7a and Supplementary Fig. 6a). siRNA for the glutathionylation enzyme GSTP compromised Act.D-induced p53 accumulation (Supplementary Fig. 6b). Consistent with

previous reports[16], the knockdown of NPM1 mRNA levels greatly reduced the accumulation of p53 following the same Act.D treatment or a short-term H$_2$O$_2$ treatment (Fig. 7b). We then added back the WT and the mutant C275S to these NPM1 knockdown cells and found that WT, but not translocation-disabled mutant C275S, could rescue the lost p53 accumulation under stress (Fig. 7c and Supplementary Fig. 6c). The levels of p53 accumulation were also demonstrated by western blotting in NPM1 knockdown and restored cells that were exposed to

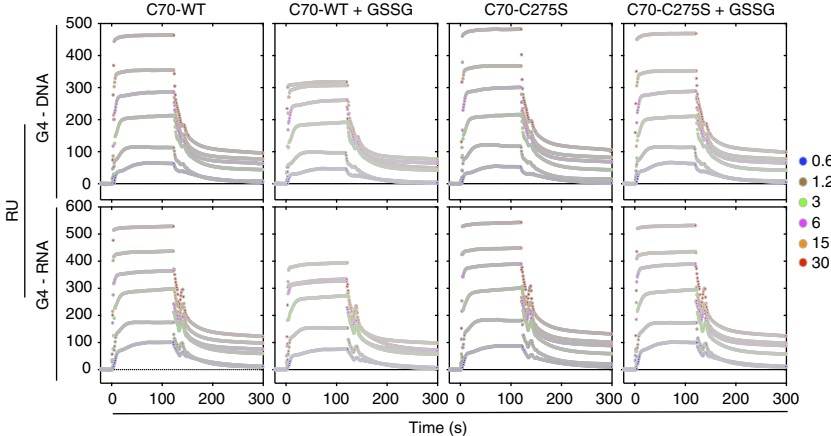

**Figure 6 | NPM1 dissociates from rRNA/rDNA after exposed to oxidized GSH *in vitro*.** SPR analysis of the interaction of NPM1-C70 (WT or C275S, as analyte) and G-quadruplexes (rDNA (upper) or rRNA (bottom), as ligand). Different colourful traces referred to increasing analyte concentrations ($\mu g\,ml^{-1}$). The dissociation constant ($K_D$) values were calculated with a one-site steady state binding model (see Supplementary Fig. 5g and Table 1). GSSG, oxidized GSH. G4, G-quadruplex.

**Table 1 | The $K_D$ values of the interactions of rRNA or rDNA G-quadruplexes with NPM1-C70 WT or C275S mutants respectively.**

| Ligand (oligo) | Analyte (NPM1) | $K_D$ ($\mu g\,ml^{-1}$) | |
|---|---|---|---|
| | | **Vehicle** | **GSSG** |
| G4-DNA | C70-WT | 5.514 ± 2.018 | 1.824 ± 0.297 |
| | C70-C275S | 4.988 ± 1.530 | 5.684 ± 1.307 |
| G4-RNA | C70-WT | 3.158 ± 1.038 | 0.8101 ± 0.574 |
| | C70-C275S | 2.613 ± 0.875 | 3.112 ± 1.049 |

G4, G-quadruplex, GSSG, oxidized glutathione.

Act.D and $H_2O_2$. Nutlin-3, an agent that directly disrupts the p53–HDM2 interaction[40], was used here as a positive control. Interestingly, Act.D or $H_2O_2$ did not produce p53 accumulation in cells bearing C275S NPM1, whereas Nutlin-3 did (Fig. 7d), indicating that unlike Nutlin-3, Act.D or $H_2O_2$-induced p53 accumulation was dependent on NPM1 translocation. Furthermore, the transcriptional activity of p53 in these cells was determined by measuring the mRNA levels of the p53 target genes *p21* and *PUMA*. The results similarly showed that p53 transactivation was impaired in the cells with C275S NPM1 (Fig. 7e). These results strongly argue that the translocation of NPM1, mediated by *S*-glutathionylation on Cys[275], is required for the activation of p53 under stress.

To clarify whether merely the nucleoplasmic localization of NPM1 could induce p53 accumulation, we repeated the rescue assays in cells with NPM1 knocked down and then with the adding back of the FLAG-tagged W288/290A mutant. Surprisingly, the W288/290A mutant, although automatically localized to the nucleoplasm, could stabilize p53 only under Act.D or $H_2O_2$ exposure conditions (Fig. 7c and Supplementary Fig. 6d). These data indicate that the nucleoplasmic localization of NPM1 is necessary but not sufficient to induce p53 activation; some putative stress-mediated modifications to p53 might be simultaneously required, which would inevitably occur under stress conditions.

Human double minute 2 homologue (HDM2) is a negative regulator of p53. Studies have revealed that NPM1 binding to HDM2 could induce p53 accumulation by blocking the activity of HDM2 E3 ligase[16]. Other papers have reported that ribosomal proteins, including L5, L11 and L23, are inhibitors of the HDM2–

p53 interaction and inducers of p53 stabilization[41–43]. We performed co-IP using an antibody against HDM2 to assess the interaction of endogenous HDM2 with exogenous FLAG-NPM1 in cells with endogenous NPM1 knocked down and the adding back of FLAG-NPM1 WT, W288/290A and C275S mutants following Act.D treatment. We found that the WT and W288/290A mutant, but not the C275S mutant, could bind with HDM2 and that the p53–HDM2 interaction was prevented only by the WT and WW mutant. In all three circumstances, ribosomal protein L23 remained bound to HDM2 (Fig. 8a). This means that although L23 is required for p53 stabilization under stress, as previous studies have shown, only NPM1 translocation to the nucleoplasm dictates the final outcome of p53 accumulation, which occurs through NPM1's binding with HDM2 and thus the disruption of the HDM2–p53 interaction.

The tumour suppressor protein ARF is a molecule that has a key function in the p53-HDM2-NPM1 interrelationship[44,45]. We next tried to assess the correlation of NPM1 translocation and ARF in respect of the activation of p53. The vector HA-p14/ARF[46] was ectopically expressed in U2OS cells that are negative in ARF expression. Examining the localization of the over-expressed ARF and endogenous NPM1 by immunofluorescent staining, we found that exogenous ARF was distributed both in the nucleolus and nucleoplasm (Supplementary Fig. 6e); the nucleolar ARF did not move out following Act.D treatment and, in contrast, the nucleolus/nucleoplasm ratio of ARF was slightly increased, whereas NPM1 translocate to the nucleoplasm under the same condition (Fig. 8b). We determined the levels of p53 by western blottings and found that, compared with the mock DNA, overexpression of ARF in U2OS cells with either normal or silenced NPM1 led to an increased accumulation of p53 under non-stress condition. However, after cells were exposed to Act.D, overexpression of ARF alone did not produce further accumulation of p53 in NPM1 knockdown cells. An enhanced p53 accumulation was observed when WT NPM1 was added back, but adding back of C275S mutant could not causes the increase of p53 (Fig. 8c). These data suggest that the nucleoplasmic fraction of ARF alone is able to induce p53 accumulation under basal conditions, yet the further accumulation under stress conditions is determined by the presence of the nucleoplasmic NPM1, independent on ARF. This idea was substantiated by the results of co-IP, in which we investigated the interactions between HDM2 and p53, the added-back WT or mutant NPM1 and exogenous ARF, respectively, following Act.D

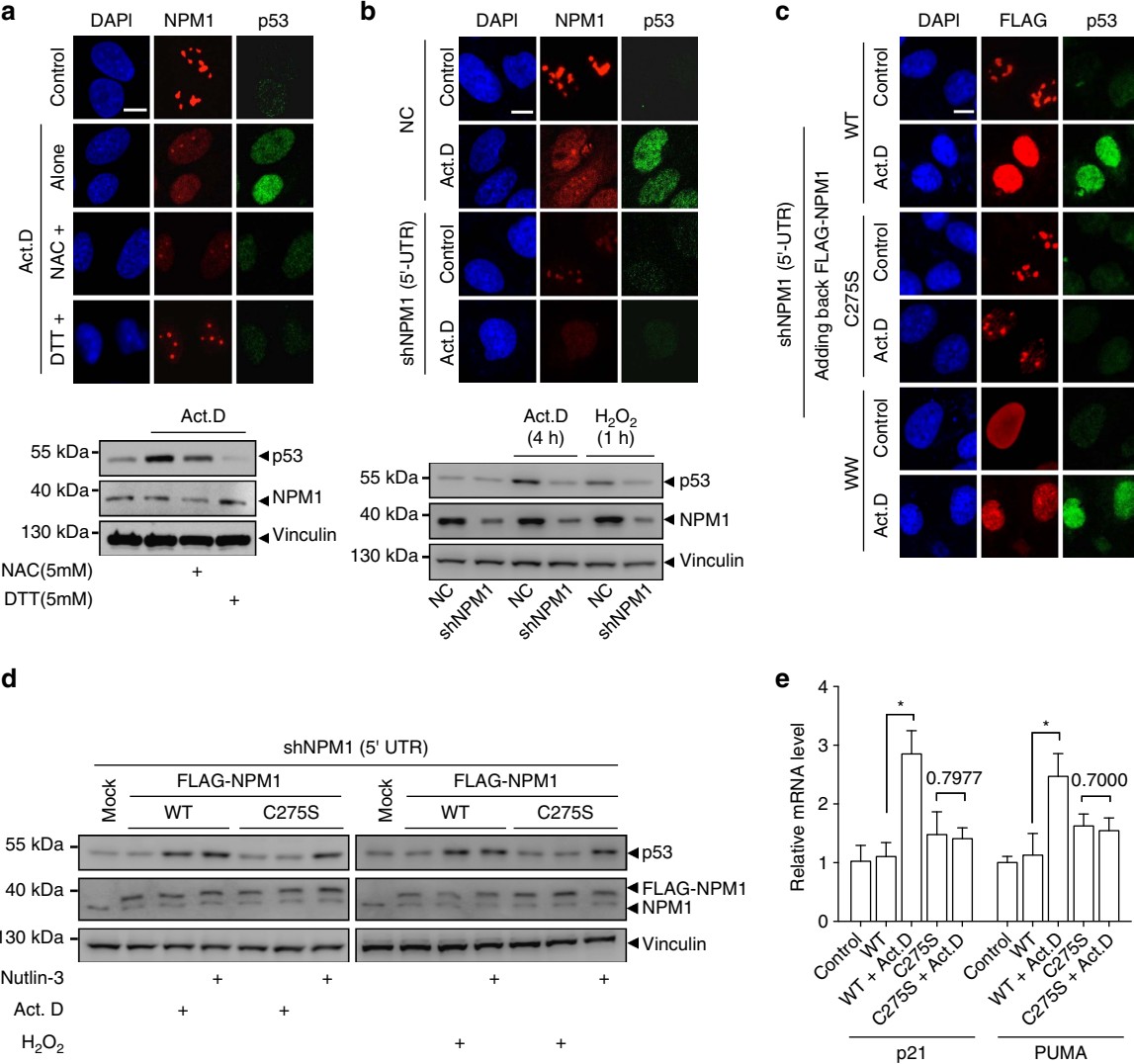

**Figure 7 | NPM1 translocation is required for p53 activation.** (**a**,**b**) Endogenous NPM1 translocation and p53 accumulation cells under nucleolar stress with western blot validation. (**a**) Cells were pretreated with NAC (5 mM) for 4 h before Act.D (8 nM) treatment, or co-treated with DTT (5 mM) and Act.D (8 nM). Nucleolar line profiles of NPM1 in representative cells are shown in Supplementary Fig. 6a. (**b**) NPM1 was silenced with shRNA (shNPM1 5′-untranslated region (UTR)) before Act.D treatment or $H_2O_2$ (500 μM). (**c**) Immunofluorescence for endogenous p53 accumulation in NPM1-silenced cells (shNPM1 5′-UTR) that added back FLAG-NPM1 WT, C275S or WW mutants before treatment with Act.D. Line profiles of p53 in representative cells were showed in Supplementary Fig. 6c. (**d**) p53 accumulation after Act.D (left) or $H_2O_2$ (right) treatment in NPM1-silenced cells which were added back WT or C275S mutant of NPM1. Endogenous and exogenous FLAG-NPM1 were blotted with anti-NPM1 antibody. Nutlin-3 was used as a positive control for p53 inducer. (**e**) Relative mRNA level changes of p21 and PUMA after Act.D treatments in NPM1-silenced cells (shNPM1 5′-UTR) that added back FLAG-NPM1 WT or C275S mutants. Unpaired $t$-test. *$P < 0.05$ or showed above the bars. Data were represented as mean ± s.e.m. Scale bars, 10 μm. WW, W280/290A. All of the experiments were performed in U2OS cells.

treatment. We found that the disruption of the HDM2–p53 interaction occurred in the cells bearing the WT but not C275S NPM1 and amounts of ARF bond to HDM2 appeared similar in both cell groups (Fig. 8d). This means that although ARF expression can promote p53 stabilization, in line with previous studies[47–49], the enhancement of p53 accumulation under nucleolar stress condition requires the presence of the nucleoplasmic NPM1.

Collectively, it is concluded that the nucleoplasmic translocation of NPM1 is a prerequisite for p53 stabilization under nucleolar stress condition.

In summary, redox changes in the nucleolar compartment and the consequent S-glutathionylation at $Cys^{275}$ serve as a fundamental mechanism for NPM1 sensing nucleolar and other stresses. The translocation of NPM1 to the nucleo-

plasm, which is due to its dissociation from rRNA and rDNA, accounts for the NPM1 regulation of p53 activation under stress (Fig. 9).

## Discussion

To our knowledge, the redox state of the nucleolus has not previously been addressed, although methods for analysing the redox conditions in specific subcellular compartments in living cells have been rapidly developed[50,51]. We constructed a nuclear-specific redox sensor to observe the process of nucleolar oxidation on different cellular insults. The nucleoli became prone to oxidation after cells were exposed to various insults that had been previously reported to induce nucleolar stress[7]. Thus, nucleolar stress can be signified by oxidative stress in the nucleoli.

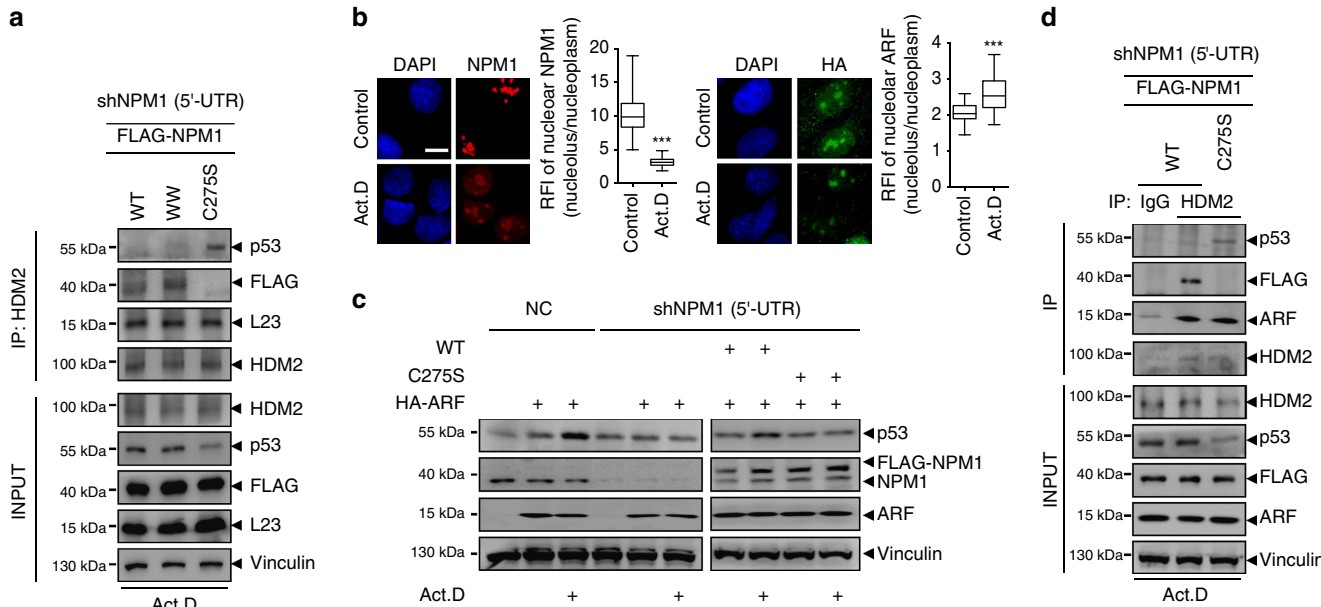

**Figure 8 | The presence of nucleoplasmic NPM1 is a prerequisite for stress-induced p53 activation.** (**a**) FLAG-NPM1 WT and the mutants WW and C275S were added back to NPM1-silenced cells before treatment with Act.D. Co-IP assays were performed using antibody against HDM2 and western blottings were performed using antibodies as indicated. (**b**) Localization of exogenous HA-ARF and endogenous NPM1 after Act.D treatment, from respective cells on same coverslip. Nucleolar/nucleoplasmic RFI ratio of NPM1 ($n = 29$) and ARF ($n = 41$) were displayed. Mean ± s.e.m. Unpaired $t$-test. ***$P < 0.001$. (**c**) p53 accumulation in HA-ARF-transfected normal and NPM1-silenced cells (left) or in FLAG-NPM1 WT or the mutant C275S added back NPM1-silenced cells (right), with or without Act.D treatment as indicated. (**d**) HA-ARF was co-transfected with FLAG-NPM1 WT or the mutant C275S in NPM1-silenced cells before treatment with Act.D. Co-IP assays were performed using IgG or antibody against HDM2, and western blottings were performed using antibodies as indicated. Scale bars, 10 μm. WW, W280/290A. All of the experiments were performed in U2OS cells.

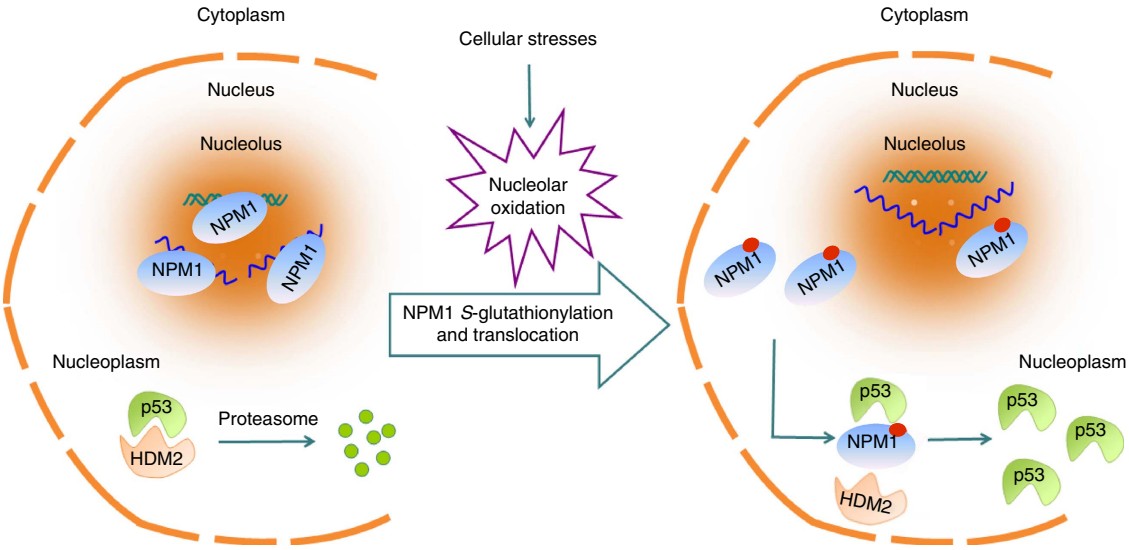

**Figure 9 | NPM1 sensing for nucleolar stress.** Nucleolar oxidation is a general response to various cellular stresses. During nucleolar oxidation, NPM1 undergoes S-glutathionylation on Cys$^{275}$, which triggers the dissociation of NPM1 from nucleolar nucleic acids. The nucleoplasmic NPM1 is indispensible for p53 activation induced by nucleolar stress.

Translocation of the protein NPM1 from nucleoli to the nucleoplasm was initially observed when ribosome biogenesis was halted by Act.D[52], and soon thereafter was also found in cells exposed to cytotoxic agents[14], viral proteins[53], ultraviolet radiation[17], heat shock[15] and agents inducing DNA damage[54], apoptosis or senescence[55,56]. NPM1 is thus considered to be a sensing molecule for stress[10,57]. Why cellular stressors that differ in nature all lead to NPM1 translocation is puzzling. Here, our findings provide an explanation by showing that a redox change

in the nucleolar compartment is a general response to diverse stress conditions.

NPM1 is believed to bind to rRNA[21,58,59]. The capability of NPM1 to bind to DNA in the nucleoplasm has also been reported[60]; nevertheless, only a few documents have suggested that NPM1 binds to rDNA[35,38,39]. Our RIP and ChIP results demonstrated that the associations between NPM1 and rRNA or rDNA, whether direct or indirect, were weakened by oxidative modifications. In parallel, NPM1 could be driven out of the

nucleoli by digestion with RNase or, to a lesser extent, with DNase, probably dissociating from the fibrillar centre (rDNA), fibrillar components (rRNA) and granular components (pre-ribosomal particles). Most importantly, the $S$-glutathionylation of NPM1 interfered with its binding to G-quadruplexes of rDNA or rRNA, as shown by our *in vitro* interaction assays. These findings indicate that the anchoring of NPM1 to rRNA and rDNA could be the force holding NPM1 within an unstressed nucleolus; the translocation of NPM1 under stress is due to its dissociation from rRNA and rDNA.

How NPM1 and other nucleolar proteins shuttle rapidly between the nucleolar-bound and unbound states remains an open question[3]. Based on the findings in this study, we assume that a rapid, reversible $S$-glutathionylation at $Cys^{275}$ could determine the localization of NPM1, which may also hold true in its shuttling under conditions that are considered to be unstressed. Three lines of evidence support this hypothesis. First, the nucleolar localization of NPM1 that was disrupted by a low dose of $H_2O_2$ could be rapidly auto-restored (Fig. 2c), meaning that two modification states could be switched along with physiological ROS fluctuations. Second, the nucleoplasmic translocalization of NPM1 can be mildly prevented by NAC, a GSH synthesis precursor, and be thoroughly stopped by DTT, a reducing agent that reduces the disulfide bonds of proteins, including those formed with GSSG. Third, GSTP and Grx1 could modulate the translocation dynamics of NPM1, indicating that the presence or absence of $S$-glutathionylation dictates the localization of NPM1 in the nucleoplasm or the nucleolus, respectively.

$S$-glutathionylation of NPM1 was initially reported following exposure of alkylating agent[61]. In our study, $S$-glutathionylation is eventually verified as the type of thiol modifications on $Cys^{275}$ of NPM1 following nucleolar oxidation, which is based on applications of the enzymes that specifically catalyse $S$-glutathionylation. It is worth noting that Grx is traditionally considered an enzyme catalysing the reduction of $S$-glutathionylated proteins, but is now also considered an efficient catalyst for protein oxidation, as demonstrated by assays using roGFP/rxYFP probes[62]. EGFP-Grx1-NPM1 we constructed shows an earlier and more thorough dispersion with a short recovery, which is dependent on the catalytical activity of Grx1 (Fig. 4d), confirming that Grx is responsible for a reversible $S$-glutathionylation of NPM1.

The nucleolus is considered to be a central hub coordinating the stress response[8]. How the signals of nucleolar stress can be connected to p53 stabilization, the initial step of activation of p53 pathway, is emerging as a research focus[7,11–13]. The translocation of NPM1 to the nucleoplasm occurs before p53 accumulation under ultraviolet radiation stress[12]. Notably, it was demonstrated a decade ago that NPM1 can, via its C-terminal domain, interact with p53 (ref. 17). NPM1 is also found to bind with HDM2, competing with p53 and blocking the HDM2-mediated ubiquitination of p53 (ref. 16). The interaction of ribosomal proteins with HDM2 is required for the activation of p53 induced by nucleolar stress[41–43]. ARF can also bind to HDM2, to stabilize p53 in diverse cellular events[48,63–65]. Our findings emphasize that the nucleoplasmic localization of NPM1 alone allows it to induce p53 stabilization under stress. The binding of ribosomal proteins or ARF with HDM2, which had been thought to be sufficient for p53 stabilization[41–43,48], is actually insufficient when NPM1 stays within the nucleoli.

There existed complicated reciprocal regulations between ARF and NPM1 (refs 66–70). Moreover, ARF-NPM1-p53 has often been considered as an axis that functions under varying physiological and pathological scenarios[66,71]. However, it has not been adequately studied whether ARF translocate from the nucleolus to the nucleoplasm under stress conditions and whether this translocation is required for NPM1 action on HDM2–p53 (refs 72,73). Moulin *et al.*[70] showed that depletion of NPM had little impact on the amount of ARF that associated with MDM2 and p53, which, together with their earlier report[49], indicate that only a nucleoplasmic fraction of the total ARF is involved in the interaction with MDM2. In the present study we further clarify that, unlike NPM1, ARF does not translocate to the nucleoplasm following a typical nucleolar stress. Moreover, under stress condition, the amounts of ARF bound to HDM2 in cells bearing the WT or mutant NPM1 remain similar. Therefore, irrespective of ARF, nucleoplasmic translocalization of NPM1 plays an indispensable role that connects nucleolar stress to p53 activation.

The findings in this and future studies could help with the development of therapeutic strategies[74–76], because NPM1 mutation is a founder genetic lesion in acute myeloid leukaemia (AML) and an attractive target for intervention[76–81]. It has been found that the p53-dependent death pathway is activated in AML cells carrying WT NPM1 but not in cells carrying aberrant cytoplasmic NPM1 (refs 28,82). We speculate that the abovementioned mechanism could account for the inactivation of p53 in these cells; the mislocalization of NPM1 resulted in the loss of the ability to sense nucleolar stress and to access to HDM2 in the nucleoplasm. NPM1 mutations are always heterozygous[77] and it is hypothesized that the NPM1-mutated AML cells might be vulnerable to drugs that trigger a nucleolar stress response, because it contains a low level of non-mutant NPM1 and because p53 in these cells is not mutated[78]. A few NPM1-interacting antiproliferative compounds have been identified[76,79–81], among which the natural product avrainvillamide binds to NPM1 specifically at $Cys^{275}$. Interestingly, avrainvillamide at a low dose disperses NPM1 into the nucleoplasm in the NPM1-WT AML cells but re-localizes the mutated NPM1 to the nucleoli and, at higher doses, induces apoptosis in NPM1-mutated AML cells[76,83]. More excitingly, a recent clinical application of Act.D at a concentration higher than the one used in the present *in vitro* study led to a morphologically and molecularly complete remission of an AML patient[78]. In this case, the disruption of nucleolar localization of the WT NPM1 of the patient by Act.D, probably through $S$-glutathionylation on $Cys^{275}$, might be one of the relevant mechanisms.

## Methods

**Antibodies and reagents.** Antibodies against NPM1 (60096-1-Ig) were obtained from ProteinTech Group (USA). Antibody against IgG1 (5415) was from Cell Signaling Technology. Anti-Flag M2 (F1804) and anti-tubulin (D66; T0198) mouse antibodies were from Sigma. Antibodies against vinculin (ab129002), GFP (ab290), L23 (ab112587), GSH (D8; ab19534), Fibrillarin (ab5821), p53 (E47; ab32509) and HA (ab137838) were from Abcam. Anti-p53 (Pantropic) (DO-1; OP43L) and anti-MDM2 (5B10C; OP146) were purchased from Calbiochem. GSSG (G2140), $H_2O_2$ (H1009), NAC (A9165), DTT (D0632) and Act.D (A4262) were purchased from Sigma-Aldrich. Nutlin-3 (3984) was obtained from Tocris Bioscience.

**Cell culture and treatments.** HEK293 (derived from ATCC), HEK293T, 293FT (derived from Invitrogen), HeLa and MKN45 (or U2OS (derived from ATCC)) cells were maintained at 37 °C in a humidified atmosphere of 5% $CO_2$ with DMEM high-glucose (or α-MEM) supplemented with 10% fetal bovine serum and penicillin/streptomycin. All cell lines were tested for mycoplasma contamination and experiments were conducted under mycoplasma-negative conditions. The Nutlin3 and Act.D solution was prepared in dimethyl sulfoxide. The NAC was prepared with 2 M Tris-base solution to produce a final pH of 7.4. Ultraviolet treatments were performed by applying UVC radiation using a CL-1000 UV cross-linker (UVP Inc., USA). Hypoxia treatment was performed in a triple-gas incubator (Huaxi Electronic Tec., China) with an atmosphere containing 1% $O_2$ and 5% $CO_2$. Heat-shock experiments were performed by heating the cells in a 42 °C water bath for 30 min. Cell starvation was induced by changing the medium into a nutrient-free Earle's balanced salt solution (Gibco).

**Plasmids.** The pcDNA3.1-FLAG-NPM1 was a gift of Dr Benjamin Y.M. Yung (The Hong Kong Polytechnic University, Hong Kong). The mCherry-C1-H2B was

a gift of Dr Chuanmao Zhang (College of Life Sciences, Peking University, China). To construct a nucleus-targeted roGFP1, NLS-roGFP1, the threefold nuclear localization signal ($3 \times$ NLS) DPKKKRKVDPKKKRKVDPKKEKRKV was added to the C terminus and expressed in HeLa cells using a modified pEGFP-N1 as the expression vector. For tracking NPM1 proteins in living cell, full coding sequences of NPM1 were subcloned from the pcDNA3.1-FLAG-NPM1 plasmid into pEGFP-N1 and mCherry-C1 (Clontech Laboratories Inc.). For Grx1-NPM1 fusion protein, full coding sequences of Grx1 were subcloned into the upstream of the NPM1 transcription initiation site in pEGFP-N1-NPM1. Grx1(ss), as a catalytically inactive Grx1, was generated by replacing $Cys^{23}$ and $Cys^{26}$ residues with serines.

Mutagenesis was performed using the QuickChange Lightning Site-Directed Mutagenesis Kit (Agilent Technologies) following the manufacturer's instructions and using primers listed in the Supplementary Table 1.

Plasmids DNA and siRNA oligo were transfected into cells with Lipofectamine 2000 (Invitrogen) according to the manufacturer's instructions.

For GSTP siRNA sequences, siGSTP-1 (sense): 5′-GGAGGACCUCCGCUG CAAAdTdT-3′; siGSTP-2 (sense): 5′-CUCCGCUGCAAAUACAUCUdTdT-3′.

For NPM1-shRNA (5′-untranslated region) oligonucleotides, the following sequences were annealed and subcloned into the pLVX-shRNA2 lentiviral expression vector:

5′-gatccGCCTACCGTGTTTGATAAATTTCAAGAGAATTTATCAAACAC GGTAGGTTTTTTACGCGTg-3′, 5′-aattcACGCGTAAAAAACCTACCGTG TTTGATAAATTCTCTTGAAATTTATCAAACACGGTAGGCg-3′.

To establish a stable NPM1-shRNA (5′-untranslated region) U2OS cell line, ZsGreen1 co-expressing lentiviral expression vectors were transiently transfected into 293FT cells. After 72 h, the supernatants were harvested to infect U2OS cells with 10 μg ml$^{-1}$ polybrene at final concentration. Lastly, the ZsGreen1-positive cells were sorted on a FACSAria II flow cytometer (BD Biosciences).

**Immunofluorescence.** Cells grown on coverslips were fixed in 4% paraformaldehyde for 10 min before permeabilization with 0.3% Triton X-100, washed with PBS and incubated with 2% BSA containing primary antibodies against NPM1 (1: 100, PTG-lab), p53 (1: 100, Abcam), HA (1: 100, Abcam), Fibrillarin (1: 100, Abcam) and FLAG (1: 100, Sigma) at 4 °C overnight. The monolayers were then washed with PBS and incubated with appropriate secondary antibodies in PBS at 37 °C for 2 h. Alexa Fluor 488- and 555-labelled goat anti-mouse antibodies, and Alexa Fluor 555- and 633-labelled goat anti-rabbit antibodies were obtained from Molecular Probes (Invitrogen). Cells were washed again with PBS, counterstained with 4,6-diamidino-2-phenylindole (Beyotime, China) and then used for observation. Images were taken on random, but tissue areas and cells with extraordinarily strong staining were excluded for examination.

**Microscope image acquisition.** Images were acquired using the Zeiss 710 laser scanning confocal microscopy system on a Zeiss Axio Observer Z1 inverted microscope, equipped with a Plan Apochromat $\times 63.0$, 1.4 numerical aperture oil-immersion, differential interference contrast objective and a 488 nm argon laser. For immunofluorescence assays, scanning was performed using $1,024 \times 1,024$ format, 1.58 μs pixel dwell, 12 bit depth, $4 \times$ line average and 3.0 μm pinhole. For live-cell imaging, images were recorded every minute. Cells were plated on a 35 mm glass-bottom dish (Nest Biotechnology) and observed at 12–24 h post transfection in a humidified atmosphere using a $CO_2$ incubator (PeCon) at 37 °C, scanned in $512 \times 512$ format, 0.5 μs pixel dwell, 12-bit depth, $2 \times$ line average and 8.4 μm pinhole. For redox probe observation, NLS-roGFP1 were excited with the 405 nm and 488 nm lasers, and emissions were detected at 500–550 nm, scanning with $512 \times 512$ format, 0.5 μs pixel dwell, 12 bit depth, $2 \times$ line average and 8.4 μm pinhole. Images were taken on random, but cells with extraordinarily strong or low expression levels were excluded for examination.

**Image processing and fluorescence signal quantifications.** Fluorescence images obtained from immunofluorescence and live-cell imaging were processed in ImageJ software (version 1.49 v with 64 bit Java Platform; NIH) and exported as TIFF mode files in red/green/blue (RGB) channels. To normalize individual cell differences, in general 30 cells or more in each group were examined for quantification of average fluorescent intensity. The absolute fluorescence intensity and line plots were obtained from bit channel files before the images were transferred to RGB channel files. RFI mean absolute fluorescence intensity changes relative to control groups in immunofluorescence or initiation time (0 s) in live-cell imaging. For NLS-roGFP1 images, the pixel-by-pixel ratio of the 488 nm excitation image to the 405 nm excitation image of the same cell was used to pseudocolour the images with ImageJ as previously described[84]. Briefly, the colour brightness was proportional to the fluorescence intensities in both channels. Shown as a calibration bar, the RGB values 255, 0, 255 (purple) represented the lowest 405/488 nm ratio, and 255, 0, 0 (red) represented the highest ratio. A higher ratio of 405/488 indicated a more oxidative state.

**Co-immunoprecipitation.** Cells grown at $10^7$ were washed with PBS and scraped before being suspended in 1 ml PBS on ice. After centrifugation, the cell pellet was resuspended in 1 ml lysis buffer (50 mM Tris-HCl pH 7.4, 1% Triton X-100, 150 mM NaCl, 1 mM EDTA, plus cocktail inhibitor (Roche)) for 30 min gentle

agitation at 4 °C, followed by sonication for two bursts of 10 s each at half power with a 2 mm stepped microtip (Sonics VCX-130, USA). After centrifugation, the supernatant was mixed with anti-NPM1 antibody (1: 200, PTG-lab), anti-GSH antibody (1: 100, Abcam) and anti-HDM2 antibody (1: 100, Calbiochem) overnight at 4 °C. Next, 30 μl of Protein A/G-Agarose (EMD Millipore) was added for a further 4 h agitation at 4 °C. Lastly, the resins were washed six times with 1 ml lysis buffer and then mixed with $2 \times$ SDS sample buffer, boiled 10 min and analysed by western blotting. Specifically for glutathionylated NPM1 samples, β-mercaptoethanol was not included in the $2 \times$ SDS sample buffer.

**Western blotting.** Cells were lysed in sample solution. Proteins were separated with 10% SDS–PAGE gels and transferred to nitrocellulose membranes. To reduce nonspecific background, membranes were blocked with 5% milk in TBS-T buffer. Bands were detected using various antibodies against NPM1 (1: 2000, PTG-lab), GSH (1:1,000, Abcam), L23 (1:1,000, Abcam), p53 (1:1,000, Abcam or Calbiochem), vinculin (1:2,000, Abcam), HDM2 (1:1,000, Calbiochem) and FLAG (1:1,000, Sigma) at 4 °C overnight. The membranes were incubated with peroxidase-conjugated secondary antibodies (1:5,000, Jackson) for 2 h at 37 °C before detection using ECL system (EMD Millipore). Images scanned with ImageQuant LAS 4000 mini (GE) were exported as TIFF images in RGB mode. All the uncropped versions of images were showed in Supplementary Fig. 7.

**Mass spectrometry analysis.** HEK293T cells transfected with FLAG-NPM1 were treated with 500 μM $H_2O_2$ for 30 min. FLAG-NPM1 was then immunoprecipitated with anti-FLAG M2 gel (Sigma), mixed with $2 \times$ SDS sample buffer without β-mercaptoethanol and loaded for SDS–PAGE. After examination of the Coomassie Brilliant Blue staining, the gel bands containing FLAG-NPM1 protein were clipped out and cut into small pieces in a 1.5 ml microtube. The chopped gels were washed three times with 50 mM $NH_4HCO_3$ in 30% acetonitrile by shaking at room temperature (RT) for 20 min. The gels were further incubated with 300 μl of acetonitrile at RT for 10 min. After removing the acetonitrile, the gels were dried and trypsin-digested following the manufacturer's protocol. Peptides were extracted with 85% acetonitrile and 0.1% trifluoroacetic acid for twice. After removing the pieces of gels, the remained solution was concentrated with a SpeedVac concentrator. Then, the peptides were re-dissolved in 30 μl of 0.1% formic acid for liquid chromatography–tandem mass spectrometry analysis (LTQ Orbitrap Elite, Thermo Scientific). The fragment ions observed in the mass spectra were analysed with Mascot.

**Structure model.** NPM1-GSH complex was modelled in COOT software[85] and images were prepared in PyMOL software (DeLano Scientific, http://www.pymol.org).

**RNase A and DNase I digestion.** HeLa cells grown on coverslips were permeabilized with 0.1% Triton X-100 in PBS for 2 min, washed immediately and treated with RNase A (EN0531, 1 mg ml$^{-1}$, Fermentas, Thermo) or DNase I (EN0523, 0.5 U μl$^{-1}$, Fermentas, Thermo) in solution buffer for 10 min at 37 °C and cells were then fixed with 4% paraformaldehyde for 10 min and analysed by immunofluorescence.

**RNA immunoprecipitation.** HEK293T cells were scraped into 1 ml PBS. After centrifugation at $600\,g$ for 2 min, the cell pellet was resuspended in 1 ml diethyl pyrocarbonate (DEPC)-treated lysis buffer (50 mM Tris-HCl pH 7.4, 1% Triton X-100, 150 mM NaCl, 1 mM EDTA, plus cocktail inhibitor (Roche)) containing 40 U ml$^{-1}$ Ribonuclease Inhibitor (TaKaRa, China) for 30 min agitation. Then, the samples were sonicated for two bursts of 10 s each at half power and centrifuged at 12,000 r.p.m. for 20 min to remove the debris. For the INPUT sample, 10% of the supernatant was used for western blot analysis and 10% was used for RNA extraction with Trizol reagent (Invitrogen). The rest of the supernatant was incubated with 20 μl anti-FLAG M2 gel for overnight rotation. After being washed with 1 ml diethyl pyrocarbonate-treated lysis buffer six times, immunoprecipitates were then separately subjected to western blotting or RNA extraction as described above. Samples comprising 10% of the resins were mixed with $2 \times$ SDS sample buffer and analysed by western blotting. Trizol (1 ml) was directly added to the rest of the resins for isolating the RNA. All steps were performed at 2–8 °C.

**Nuclear preparation and ChIP analysis.** HEK293T cells were cross-linked with formaldehyde (0.25%, final concentration) for 10 min at RT in dishes, then washed with PBS before being scraped into 1 ml PBS. After centrifugation, cell pellet was resuspended in 1 ml of buffer A (10 mM Hepes-KOH pH 7.4, 10 mM KCl, 1.5 mM $MgCl_2$, 0.5 mM EDTA, 0.5 mM EGTA, plus cocktail inhibitor (Roche)) and flushed through a 23 G needle syringe 27 times. The released nuclei were monitored microscopically and washed once with buffer A with centrifugation. Then, the nuclei were resuspended in 0.1 ml of TE buffer (20 mM Tris-HCl pH 7.4 and 2 mM EDTA) containing 2% SDS and incubated at 37 °C for 15 min to disrupt the nucleolar structure. An additional 0.9 ml lysis buffer (50 mM Tris-HCl pH 7.4, 1%

Triton X-100, 150 mM NaCl, 1 mM EDTA, plus cocktail inhibitor (Roche)) was added to each sample before sonication for four bursts of 15 s each at 80% power. After centrifugation at 12,000 r.p.m. for 20 min to remove the debris, 10% of the nuclear chromatin supernatant was used as INPUT for western blotting and 10% was used for genomic DNA extraction. Extraction was performed using the TIA-Namp Genomic DNA Kit (TIANGEN BIOTECH, China). Lastly, the rest of the supernatant was used immediately in ChIP assays. The nuclear chromatin super-natant was then incubated with 20 μl anti-FLAG M2 gel and rotated at 4 °C overnight. The resins were washed as follows: twice with 1 ml of lysis buffer containing 0.2% SDS, twice with 1 ml of lysis buffer and twice with 1 ml of TE buffer. Then, 10% of each sample was mixed with $2 \times$ SDS sample buffer, boiled for 10 min and analysed by western blotting, and the rest of the resins were subjected to genomic DNA extraction as described above. The purified DNA was resuspended in ddH$_2$O for real-time quantitative PCR analysis.

**Real-time quantitative PCR analysis.** Real-time quantitative PCR for RIP and ChIP assays was performed using FastStart SYBR-Green/Rox (Roche) on ABI-7500 Fast equipment (Applied Biosystems). Samples were analysed in triplicate and normalized against vehicle group as an internal control. Relative changes in gene expression were calculated using the $\Delta\Delta$Ct method. For rRNA samples, an equal volume of isolated RNA was reverse-transcribed with AMV Reverse Transcriptase (Promega). The amplification procedure was performed as follows: 95 °C 10 s, 60 °C 30 s and 72 °C 30 s for 40 cycles, with a melting-curve analysis to verify the specificity of the amplification. The raw data were exported and analysed with GraphPad Prism software. The sequences of primers used in RIP and ChIP assay are shown in the Supplementary table 1.

**Prokaryotic protein expression and purification.** The gene fragments encoding the C terminus of NPM1 (NPM1-C70) WT and C275S were fused with SUMO tag and inserted into pCDFDuet vector (Novagen) at the NcoI/XhoI restriction sites. Plasmids were transformed into *Escherichia coli* BL21 (DE3). Cells were grown in Luria broth supplemented with 50 μg ml$^{-1}$ streptomycin, induced with 1 mM isopropyl-D-thiogalactopyranoside and then transferred to 18 °C for 24 h before harvesting. The cell pellets were resuspended in Buffer A (500 mM NaCl, 20 mM phosphate and 20 mM imidazole pH 7.4) and sonicated. After separating the debris by centrifugation, the supernatant was loaded on a Ni column pre-equilibrated with Buffer A. After washing with five column volumes of wash buffer (Buffer A contain 50 mM imidazole), the His-SUMO-tagged protein was eluted with Buffer B (500 mM NaCl, 20 mM phosphate and 500 mM imidazole pH 7.4). Fractions containing the His-SUMO-tagged protein were then desalted and exchanged to digestion buffer (20 mM Tris-HCl pH 8.0, 150 mM NaCl and 1 mM DTT) and digested with SUMO protease at 37 °C for 1 h. The cleavage mixture obtained was then buffer changed to Buffer A (500 mM NaCl, 20 mM phosphate and 10 mM imidazole pH 7.4) and loaded onto Ni column to separate WT or C275S from the His-SUMO and the uncleaved fusion protein. At last, WT or C275S was recovered from the flow through and buffer exchanged to 50 mM Tris-HCl pH 7.4 by ultrafiltration, concentrated and ready to use.

**Surface plasmon resonance.** Oligonucleotides (oliogos) for SPR analysis were prepared following the study of Chiarella *et al.*[35]. One of a G-quadruplex-forming sequences in 47S rRNA precursor, 5′-GGGGTGGGGGGGGAGGG-3′, namely 13079NT, was selected to represent the rDNA nucleic acid and its rRNA counterpart was selected to represent the rRNA nucleic acid. Lyophilized HPLC-grade DNA or RNA oligos were obtained from Invitrogen. The two oligos were both biotinylated at their 5′-end and dissolved in 20 mM Hepes pH 7.4 with 150 mM KCl. For annealing to form G-quadruplex structure, both oligos were heated at 95 °C for 15 min and were let to gently cool down overnight at room temperature.

SPR measurement was performed on a BIAcore T200 instrument with four flow channels and a sensor chip with dextran matrix and pre-coated streptavidin (GE Healthcare). The chip was activated with three consecutive 1 min injections of 50 mM NaOH and 1 M NaCl. For immobilization of G-quadruplex oligos, the flow rate was 5 μl min$^{-1}$ to minimize sample consumption. Biotinylated DNA (10 μM) and RNA oligos diluted with the running buffer was injected and captured onto the channel surface, to achieve the immobilization level at about 300 RU (response unit) of both. For the binding assay, the reaction temperature was controlled at $25 \pm 0.1$ °C and the flow rate was set at 30 μl min$^{-1}$. The response obtained from the detection channel (FcDNA or FcRNA) was normalized by subtracting the signal simultaneously acquired from the control channel (Fcvehicle), which could eliminate nonspecific binding and buffer-induced bulk refractive index changes. Tris-HCl (50 mM, pH 7.4) was used as running buffer. Different concentrations (0.6, 1.2, 3, 6, 15 and 30 μg ml$^{-1}$) of WT and C275S of NPM1-C70 were prepared and their bindings with rDNA or rRNA were measured in SPR assay. To specify the effect of GSSG on NPM1-C70/G-quadruplex binding, some of NPM1-C70 samples were mixed with 10 mM GSSG. The sample solutions were injected onto the chip surface for 120 s. After each binding reaction, a further dissociation time of 180 s was applied and as the regeneration buffer, 10 mM NaOH solution was injected for 60 s to allow the signal back to the baseline.

Owing to the fast $K_{on}$ and $K_{off}$, the $K_D$ values were calculated with a one-site steady-state binding model[86], with equation $Y = B_{max} \cdot X/(K_D + X) + \text{background}$, where $X$ is different analyte concentration and $Y$ is the corresponding maximal response unit.

**Statistical analysis.** GraphPad Prism software (version 6.0c) was used to analyse and plot all data. Statistical analysis was performed with two-tailed unpaired *t*-test with 95% confidence interval. In box and whisker plots (Figs 1b and 2a), the box showed the top and bottom quartiles (25–75%) with a line at the median and the whiskers showed the minimum to the maximum values of all data. *P*-values indicated the significances of differences.

**Data availability.** The data that supports the conclusions of this study are available from the corresponding author upon request.

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

## Acknowledgements

We thank Drs Benjamin Y.M. Yung of The Hong Kong Polytechnic University, Chuanmao Zhang of Peking University and Jim Remington of Oregon State University for their gifts of the plasmids, the pcDNA3.1-FLAG-B23 and the mCherry-C1-H2B and roGFP1, respectively. This work was supported by grants from the National Natural Science Foundation of China (31230037 and 31471263), the Ministry of Science and Technology of China (2013CB910900) and Shanghai Municipal Science and Technology Commission 11JC1406900, 11DZ2260200, 16ZR1418400 and 15YF1402600, the Lift Engineering for Young Talent of China Association for Science and Technology (to Y.Z.).

## Author contributions

K.Y., J. Yi, J. Yang and Y.Y. designed experiments. K.Y., M.W., Y.Z., X.S., X.L. and H.C. conducted experiments. A.Z. analysed structure and mutation. H.Z. analysed mass spectrum data. J.X. conducted SPR. Y.Y. and M.W. provided plasmids. K.Y. and J. Yi wrote the manuscript.

## Additional information

**Competing financial interests:** The authors declare no competing financial interests.

