## [Peer Review File · Nature Communications]

Reviewer #1 (Remarks to the Author)

B23 (also and preferably known as NPM1 or Nucleophosmin) is a major nucleolar phosphoprotein and a key chaperone implicated in the ribosomal stress response. B23 functions for example in ribosome biogenesis, histone dynamics, protein shuttling, centrosome duplication, DNA replication and repair. Under normal growth conditions B23 is predominantly localized to the nucleolus but upon cellular stress there is a shift in B23 localization from the nucleolus to the nucleoplasm.

In this manuscript the authors investigated the possibility that redox changes in the nucleolar compartment affected B23 post translation modification patterns and the functional role of such changes in B23 localization dynamics and control of p53.

The present study links stress induced oxidative changes in the nucleolus to an increase in glutathionylation of B23, to the translocation of B23 out of the nucleolus, and finally to the control of the ribosomal stress response and p53 activation. The findings have some novelty and I do find some of them interesting and of potential importance. The novel findings are: (1) modification of B23 occurs at amino acid residue C275 by S-glutathionylation; (2) there are profound redox changes in the nucleolus following some types of cellular stress; (3) the modification of C275 affects B23 translocation and regulation of the p53 tumor suppressor; and in support (4) NAC (N-acetylcysteine) has a major effect on the ribosomal stress response.

My main critique is that the study relies on results from one cell line, HeLa, and we do not know how general the observed effects are. Does nucleolar oxidation occur in other cell types including normal cells, and if so, is B23 modified?

There are some additional issues that need to be further clarified and examined.

1. The first thing that I reacted on is the nucleolar enrichment of the reporter (Figure 1). Does this redox reporter GFP molecule shuttle between the nucleoplasm and the nucleolus? What I hint at is whether the reporter actually reacts more on the redox state in the nucleoplasm than in the nucleolus. How fast does the reporter sense a redox change in relation to any potential rapid shuttling? The nucleolus is not a membrane confined compartment such as ER or mitochondria.
2. Does the C275S mutant heteroligomerize (dimerize) with wt B23? More interestingly, does B23 modified at C275 with glutathione heterodimerize with wt B23? In order to simplify the experimental approach it would probably be sufficient to conduct a co-IP experiment.
3. Figure 4A. The experiment is using wild type B23. The same experiment should be repeated but with the B23 mutant C275S as comparison.
4. Figure 5-6. It seems as if the transfection efficiency is poor. Is that a reflection of few cells transfected or a low expression of FLAG-B23 in general but many cells transfected? Another critical issue is the knockdown efficiency of B23. From panels in figure 6 we can see that overexpression of FLAG-B23 is inefficient due to the large amounts of B23 in cells. Given this, one must assume that the knockdown of B23 should be efficient in order for the experiment with FLAG-B23 mutant to actually work... The authors should in the WBs in Figure 6 show the knockdown of endogenous B23 in relation to what was re-introduced back.
5. Figure 6A. Where did B23 go? In panel A I barely see any B23 in the images, but according to WB there is no major change in the levels of B23. Did the authors mix up the images with the B23 knockdown in panel B? There should be a more visible B23 signal from the nucleoplasm.
6. Figure 7. Should read glutathionylation.
7. Table 1. It is a bit unclear from the main text and even the Table 1 regarding whether it was actually full length mutant B23 C275S that was used in comparison with the B23-C70 construct. It

is imperative that the comparison is made with the same size construct, which I assume is the case by reading supplement info. This needs to be stated more clearly in the main text.

8. It would strengthen the paper to include a control for p53 activation, such as Nutlin-3. Activation of p53 by Nutlin-3 occurs by disruption of MDM2-p53 interaction and this should at least in theory be unaffected by the status or localization of B23. In such control experiment they may also want to include NAC.

9. The authors have not investigated the activation of any of the known p53 target genes such as p21 or PUMA. We therefore do not know how the activity of p53 has changed in relation to the altered levels of p53 protein.

10. Discussion: The authors argue that B23 binding to rDNA or rRNA, disrupted by RNase/DNase treatment, that is of critical importance in the translocation of B23 to the nucleoplasm. Another possibility that should be mentioned is that RNase treatment destroys pre-ribosomal particles, and this indirectly may cause B23 diffusion to the nucleoplasm. The authors hint at this, but I think it should be stated more explicitly.

11. Methods: The authors need to better describe in the material and methods (supplemental part), exactly how the RFI (relative fluorescence intensity) was measured and calculated. This is a very central aspect of the paper since many of the images suggest a very strong nucleolar signal.

12. The grammar (style) and spelling in the manuscript does not reach up to the high standard one would expect in a Nature Communication publication. The authors need to arrange for editing of the text, as some sections are a bit hard to understand at a first reading.

13. An interesting nonpeptide compound is avrainvillamide, an alkaloid, that targets residue C275 in human NPM1 and also induces p53 activity in cells, this paper published in 2007 (Wulff JE, et al JACS 2007) is included in the references list but the authors may want to discuss their results in relation to the study from 2007 even more. Such discussion is important since it concerns the same residue C275. In my opinion more important than the lengthy discussion on AML and NPM1+.

Reviewer #2 (Remarks to the Author)

There is a burgeoning literature detailing how S-glutathionylation of proteins can alter both their structure and function and that the subsequent changes underlie changes in a number of critical cellular pathways. The present communication adds to this by defining how this post-translational modification serves as a general transmitter of stress through the nucleolus. Even though the reports are not discussed in the present paper, there is precedent that S-glutathionylation of key proteins can impact stress response pathways in other organelles (such as endoplasmic reticulum or mitochondria) and consequently, there is reason to ascribe significance to this post-translational modification in the context of cell stress pathways. In general, the types of stresses used in the present report are quite broad and there is a need to understand how they might coalesce through redox changes that impact specific cysteine residues. In the same context, under the conditions used, how B23 glutathionylation might "compete" with similar modifications of other proteins, and the precise mechanism by which cysteine 275 is activated would be informative. In particular, since the authors proceed to study the impact of B23 on p53, the fact that the latter is also subject to this post-translational modification might be important to recognize and discuss.

Issues that require some specific attention include:

1. Glutaredoxin has been variously ascribed function(s) in both the forward and reverse reaction of S-glutathionylation. In the case of the glutathionylation cycle, there are enzymes that specifically

catalyze the forward reaction. As such, glutaredoxin represents a curious choice in the context of the present study.

2. A better understanding of how the various types of stress employed in the present series of experiments funnel through redox pathways might be informative. For example, actinomycin D is not generally associated with direct alterations in pathways that would be expected to lead to oxidation of cysteine residues to states that would encourage S-glutathionylation (e.g. creation of cysteinyl residues).

3. Are the authors considering that nucleolar specific pools of GSH/GSSG are critical to the modification of proteins? Does deglutathionylation of B23 occur in the nucleolus and result in its translocation back to the cytosol?

4. NAC and DTT have some degree of redox specificity for influencing GSH pools and S-glutathionylation. There may be some value in discussing their relative impact on cysteine oxidation and GSH flux.

5. p53 is also S-glutathionylated (Velu et al, Biochemistry, 2007). Is this relevant to the various protein:protein interactions studied here? Also, what is the stoichiometry of B23 with p53 and the other proteins in the complex? Is there any competitive process that might influence S-glutathionylation of each protein?

6. Translocation of B23 can also be regulated by JNK. There is literature concerning the importance of redox regulation of JNK and its impact on protein binding partners. It seems possible that such considerations may also be important here?

Reviewer #3 (Remarks to the Author)

The ms "Nucleolar stress sensing by B23 " address the causes of translocation of nucleolar protein B23 to the nucleoplasm that have been described in the literature to occur in response to many types of cellular stresses. However, it is unknown what activates the translocation. These questions are relevant given that B23/NPM1 is a protein that regulates the stabilization and activation of p53, and is also mutated in hematologic cancers. In this work, it is demonstrated that many of the nucleolar stresses alter the redox state of the cells, that these changes occur in the nucleolus, and that B23 localization changes are affected by the changes in the redox state. The authors identify that B23 C275 undergoes glutathionylation, and show that mutation of this site abrogates B23 nucleoplasmic translocation. They demonstrate, using surface plasmon resonance, that the binding of wt B23, but not C275S mutant is abrogated by oxidized glutathione (GSSG) in assays using both DNA and RNA oligomers. These findings further support the interpretation that B23 nucleolar retention is dependent on its nucleic acid interactions in a manner dependent on its oxidative state. They also address events that lead to p53 stabilization by B23, and demonstrate that the C275S mutant does not bind HDM2.

The study is well conceived and conducted and provides a convincing argument supporting B23 redox regulation. As such, it provides a significant advance in the field of nucleolar biology.

Reviewer #4 (Remarks to the Author)

Review of manuscript:

Nucleolar stress sensing by B23: a redox mechanism underlying its nucleoplasmic translocation and p53 modulation

A. Summary of the key results:

Nucleophosmin (NPM1) is a nucleolar protein involved in cellular stress response and stabilization of p53. During cellular stress NPM1 translocates from the nucleolus to the nucleoplasm and this is key to p53 stabilization. Yet how this translocation is mediated is mechanistically unclear and is the focus of this manuscript. In this manuscript the authors examine the cause of translocation of NPM1 from the nucleolus to the nucleoplasm. The authors use innovative colorimetric single cell assays and a variety of tests and mutants to provide data in support of the idea that NPM1 translocation is controlled by a redox mechanism involving cysteine 275, and that this

phenomenon could play a role in stabilization of p53.

B. Originality and interest:

The results are interesting, potentially important and to the knowledge of this reviewer, novel.

C. Data & methodology: validity of approach, quality of data, quality of presentation:

The authors' assertion that NPM1 becomes dispersed in the nucleus upon stress appears to be true for hydrogen peroxide but not that convincingly for actinomycin D.

The experiments in Figure 2 were carried out with hydrogen peroxide, again supporting the idea that this is the stressor that should be used in all experiments.

Given the importance of technique in the execution and interpretation of SPR experiments, the authors should provide more details in the methods section accompanying the actual manuscript, as opposed to in the supplementary information. The mathematical model of the SPR experiment overlaid on the curves does not fit the data well. The modeled dissociation goes to zero whereas this does not happen in the actual data. This issue should be addressed by the authors. From the online methods section it is unclear how the authors calculated the KD, since it is stated that two methods were used. The first method (using association and dissociation rates) would not provide accurate values given the fact that the modeled dissociation rates do not fit the data. The surface density coated by the investigators is too high which may affect the data quality. This is reinforced by the fact that lower concentrations of protein appear to give a better fit. A much lower surface density would still provide high enough signal to give curves that can be interpreted. It is thus recommended that the investigators coat tenfold less DNA or RNA on their sensor chips.

The authors transition to the use of actinomycin D in the end of the manuscript, at Figure 6. If stress causes p53 stabilization, then this should be observed with H₂O₂ treatment as well. All previous experiments were carried out with peroxide. Actinomycin D is highly toxic, affects the production of all proteins through its inhibition of RNA polymerase, and thus results obtained with actinomycin D can be artifactual. All experiments should at least be carried out with the same compound (H₂O₂).

The authors refer to the use of Actinomycin D in clinical trials. How does the dose used in their experiments compare with the clinically used dose?

D. Appropriate use of statistics and treatment of uncertainties:

The authors should clarify the statistical significance designation in Figure 1. For example, they should indicate whether any of the NAC-treated cells are significantly different from control NAC-treated cells. It is noted that in the presence of UV, the treated sample appears to be higher than the control sample.

In the last sentence on page 8 the authors state that the modification "apparently impeded the association of the interface with the potential nuclear macromolecules", however the data shown is a simulation and should only be stated to suggest a mechanism.

E. Conclusions: robustness, validity, reliability:

In the first full paragraph of page 6 of the manuscript the authors suggest that the observations in the presence of DTT have a strong causal implication, however these observations are merely correlations. It is suggested that this sentence be reworded.

On page 9 the authors imply their observations of cells treated with RNAs and DNAs to show a causal relationship when in fact this is an association under conditions that are highly abnormal. The text should be more carefully worded using the words "suggest" rather than "confirmed".

At the top of page 10, beginning of the paragraph, the authors should use the word "test" as opposed to "verify".

F. Suggested improvements: experiments, data for possible revision:
Please see all issues mentioned above.

In addition:

The authors should use the HUGO nomenclature NPM1 for the gene nucleophosmin.

The use of the English language by the authors could use some polishing and reviewed by a native speaker is recommended.

The authors are encouraged to use different color schemes for different treatments in Figure 5C. The use of similar colors for different treatments in the adjacent panels is confusing.

G. References: appropriate credit to previous work?

The authors appear to reference previous work correctly.

H. Clarity and context:

The abstract requires clarification in that it is unclear that the authors are referring to a different mutant (than the Cys275 one) when they discuss "the mutant protein with nucleoplasmic location".

In the discussion, the following point should be clearly addressed: How does the dose Actinomycin D dose used in their experiments compare with the clinically used dose?

Reviewer #5 (Remarks to the Author)

The manuscript entitled "Nucleolar stress sensing by B23: a redox mechanism underlying its nucleoplasmic translocation and p53 modulation" describes investigations in the role of the protein B23 in sensing nucleolar mediated stress response upon different stimuli. The Authors have used different stimuli (H₂O₂, hypoxia, UV irradiation, heat shock, starvation and Act. D) to change the cell redox state. They show that the change in redox state causes the release of B23 from the nucleolus and that this release requires a post-translational modification of cysteine 275 which, upon stress, becomes S-glutathionylated. Furthermore, the Authors show that the release of B23 from the nucleolus consistently coincides with loss of rDNA and rRNA binding. Thus, the Authors postulate that the already known ability of B23 to bind nucleic acids is at the basis of its location in the nucleolus, and that its relocation to the cytoplasm is due to S-glutathionylation of cysteine 275 upon stress, which, in turn, causes its dissociation from rRNA and rDNA, as shown by mutagenesis experiments. Finally, the Authors address the effect on p53 stabilization of the translocation of B23 to the cytoplasm, since B23 translocation and the activation of p53 as a stress response have been often described to occur together under nucleolar stress. They conclude that a stress mediated B23 nucleolar release is required for the regulation of p53 activation under stress.

The manuscript is clearly and logically presented. However, while the Authors have provided convincing and detailed data on the mechanism underlying the release of B23 from the nucleolus upon several stress stimuli, as well as on the role of S-glutathionylation in mediating the interaction of B23 with nucleic acids, the data on the ability of B23 to bind rDNA and rRNA or to interfere with the p53-HDM2 interaction upon stress dependent release, with consequent p53 accumulation, are not novel (Kurki S. et al, Cancer Cell, 2004, 5: 465-75, also cited by the Authors). Notably, the Authors show that nucleoplasmic localization of B23 is necessary but not

enough to induce p53 accumulation, indicating that others stress induced mechanisms might be involved.

A limit of this manuscript is indeed the lack of real investigations in the mechanisms of p53 stabilization by B23.

In particular:

- In Fig. 6, in order to correlate B23 localization, as well as that of the mutants, with p53 stabilization, the Authors should perform cellular fractionation to separate the nucleus from the cytoplasm and show their levels in the two compartments.
- The Authors should verify whether p53 is only phosphorylated or acetylated (posttranslational modification by which p53 stabilization and function are regulated) in the presence of B23
- The Authors should also verify p53 levels of activity evaluating the expression of its transcriptional targets
- The Authors should assess the role of ARF (which has a key function in the p53-mdm2-B23 pathway) by using a cell line expressing it, since U2OS are ARF negative
- The Authors should assess the effect of other p53 stabilizing treatments utilizing alternative oxidative stimuli (such as H₂O₂), which also induce S-glutathionylation of B23, in order to confirm the act. D data. They should also examine whether this pathway is active only in the presence of oxidative stress, by using different stimuli which do not produce oxidative stress, S-glutathionylation of B23, and/or translocation of B23 from the nucleolus, but can stabilize p53, likely through other pathways.

Minor points

- Fig. S5: The Authors use GADPH as negative control of RIP and ChIP experiments, however, we do not think this is appropriate as rDNA is present in hundreds of copies in the genome. As negative control, the Authors should use a region of rDNA or rRNA where they do not expect to find B23 binding.

The following is a point-point reply to reviewers.

Reviewer #1 (NPM1 expert):

NPM1 (also and preferably known as NPM1 or Nucleophosmin) is a major nucleolar phosphoprotein and a key chaperone implicated in the ribosomal stress response. NPM1 functions for example in ribosome biogenesis, histone dynamics, protein shuttling, centrosome duplication, DNA replication and repair. Under normal growth conditions NPM1 is predominantly localized to the nucleolus but upon cellular stress there is a shift in NPM1 localization from the nucleolus to the nucleoplasm. In this manuscript the authors investigated the possibility that redox changes in the nucleolar compartment affected NPM1 post translation modification patterns and the functional role of such changes in NPM1 localization dynamics and control of p53.

The present study links stress induced oxidative changes in the nucleolus to an increase in glutathionylation of NPM1, to the translocation of NPM1 out of the nucleolus, and finally to the control of the ribosomal stress response and p53 activation. The findings have some novelty and I do find some of them interesting and of potential importance. The novel findings are: (1) modification of NPM1 occurs at amino acid residue C275 by S-glutathionylation; (2) there are profound redox changes in the nucleolus following some types of cellular stress; (3) the modification of C275 affects NPM1 translocation and regulation of the p53 tumor suppressor; and in support (4) NAC (N-acetylcysteine) has a major effect on the ribosomal stress response.

---We thank the reviewer for this summary.

My main critique is that the study relies on results from one cell line, HeLa, and we do not know how general the observed effects are. Does nucleolar oxidation occur in other cell types including normal cells, and if so, is NPM1 modified?

---We performed three assays including nucleolar redox changes, endogenous NPM1 translocation and the glutathionylation of NPM1 in HEK293, a non-tumor cell line, U2OS, an osteosarcoma cell line and MKN45, a gastric cancer cell line. We found that after the cells were exposed to hydrogen peroxide, the nucleolar oxidation, NPM1 translocation and NPM1 glutathionylation happened in all cell types, thus confirming the generality of the effects. The data were added as **Supplementary Figure 1b, 2c, 4b.**

There are some additional issues that need to be further clarified and examined.

1. The first thing that I reacted on is the nucleolar enrichment of the reporter (Figure 1). Does this redox reporter GFP molecule shuttle between the nucleoplasm and the nucleolus? What I hint at is whether the reporter actually reacts more on the redox state in the nucleoplasm than in the nucleolus. How fast does the reporter sense a redox change in relation to any potential rapid shuttling? The nucleolus is not a membrane confined compartment such as ER or mitochondria.

---The reviewer raised an interesting question that we didn't ask previously. But after discussion with some experts working on redox probe, we remained unclear about whether these proteins might shuttle or not. As shown in the Supplementary figure 1, told by an almost

homogenous immunostaining of anti-GFP, the quantity of NLS-roGFP1 in the nucleolus is similar to that in the nucleoplasm. Based on this recognition, we speculated that the nucleolus/nucleoplasm differences in the intensity of the fluorescence reflected the differences of the redox states between two compartments. To verify this assumption we recorded the values of 405/488 ratio of the nucleoli and nucleoplasm respectively. The results showed that the ratio values in the nucleoli were higher than those in the nucleoplasm, meaning that the nucleoli are more oxidation prone (see **Figure for reviewer 1 a, b**). We have no idea about how to determine the speeds the reporter senses the redox change. In literature we can know that the redox reporters sense and report very rapidly, which might be considered as in the real-time fashion (**Dooley et al., 2004**).

2. Does the C275S mutant heterodimerize (dimerize) with wt NPM1? More interestingly, does NPM1 modified at C275 with glutathione heterodimerize with wt NPM1? In order to simplify the experimental approach it would probably be sufficient to conduct a co-IP experiment.

---We performed co-IP after cells were transfected with the WT and the mutant NPM1 labeled by two different tags. The results showed that the C275S mutant can form heterodimer with WT NPM1, as the binding of two proteins was detectable. The glutathionylation was detectable on the immunoprecipitates of two WT NPM1 with different tags, indicating that the glutathionylation dose not affect NPM1 dimer formation. The data were shown as **Figure for reviewer 1 c, d**.

3. Figure 4A. The experiment is using wild type NPM1. The same experiment should be repeated but with the NPM1 mutant C275S as comparison.

---We followed this suggestion. The results turned out to be complicated. The mutant C275S had glutathionylation as well, which could be reversed by NAC, indicating the other two cysteine residuals could be modified, although they don't involve in the NPM1 translocation. The data were shown as **Figure for reviewer 1 e**.

4. Figure 5-6. It seems as if the transfection efficiency is poor. Is that a reflection of few cells transfected or a low expression of FLAG-NPM1 in general but many cells transfected? Another critical issue is the knockdown efficiency of NPM1. From panels in figure 6 we can see that overexpression of FLAG-NPM1 is inefficient due to the large amounts of NPM1 in cells. Given this, one must assume that the knockdown of NPM1 should be efficient in order for the experiment with FLAG-NPM1 mutant to actually work. The authors should in the WBs in Figure 6 show the knockdown of endogenous NPM1 in relation to what was re-introduced back.

---We are grateful to the reviewer for these critical comments. As NPM1 is an abundant protein, the low efficiency in both overexpression and knockdown may account for insignificant effects of adding-back. A big quantity of endogenous NPM1 was remained after insufficient knockdown, plus an insufficient overexpression of the mutant; these diluted the outcomes of the adding-back. Your comments made us to input great efforts. We constructed 3xFlag tagged WT and mutant NPM1 to enhance the transcription efficiency. We also constructed a viral vector with high titration to enhance the knockdown efficiency. Based on these materials we re-performed the majority of experiments in Figure 6 (see new **Figure 6b, c, d, e** and **Supplementary Figure 6d**). The exogenous FLAG-NPM1 and endogenous NPM1

levels after knockdown/adding-back were indicated as the reviewer suggested (**Figure 6d** and **Supplementary Figure 6d**).

5. *Figure 6A. Where did NPM1 go? In panel A I barely see any NPM1 in the images, but according to WB there is no major change in the levels of NPM1. Did the authors mix up the images with the NPM1 knockdown in panel B? There should be a more visible NPM1 signal from the nucleoplasm.*

---There is indeed no major change in the total levels of NPM1 upon stress. This reviewer's questions make sense. The reason for you didn't see more visible NPM1 signal from the nucleoplasm might be that NPM1 became diluted after moved to the nucleoplasm that was relatively an immense area. What's worse, the images might be not representative enough. To solve these problems we, based on the experiments with enhanced transfection and knockdown efficiencies, reorganized the Figure 6. The new images showed line plots measuring the fluorescent intensities in the nucleoli and nucleoplasm in individual cells. The data, presented as **Supplementary Figure 2a, 6a, indicated an increased intensity in the nucleoplasm upon stress.**

6. *Figure 7. Should read glutathionylation.*

---Thanks to this critic, we corrected the misspelling.

7. *Table 1. It is a bit unclear from the main text and even the Table 1 regarding whether it was actually full length mutant NPM1 C275S that was used in comparison with the NPM1-C70 construct. It is imperative that the comparison is made with the same size construct, which I assume is the case by reading supplement info. This needs to be stated more clearly in the main text.*

---We re-wrote the table contents to make them clearer.

8. *It would strengthen the paper to include a control for p53 activation, such as Nutlin-3. Activation of p53 by Nutlin-3 occurs by disruption of MDM2-p53 interaction and this should at least in theory be unaffected by the status or localization of NPM1. In such control experiment they may also want to include NAC.*

---We really appreciate this instructive suggestion. We first tested the effects of Nutlin-3 treatment for 4 h on ROS generation and NPM1 translocation, and found that it didn't cause these two events in our system, U2OS cells. Nutlin-3 treatment for 4h could induce p53 accumulation in U2OS cells, which could not be reversed by the anti-oxidant NAC or thiol-reducing agent DTT (Figure for reviewer 1 f-h**). These data indicated that Nutlin-3 induced p53 accumulation in a manner different with that used by the nucleolar stress inducers. Finally, we use Nutlin-3 in an assay using Act.D or H₂O₂ in U2OS cells with enhanced knockdown and adding-back efficiencies. The results showed that while Act.D or H₂O₂ could not lead to p53 accumulation in cells bearing C275S NPM1, Nutlin-3 could (**Figure 6d**). This confirms this reviewer's speculation that the action of Nutlin-3 is unaffected by the status or localization of NPM1. With such control, we can conclude that, differently with Nutlin-3 that direct disrupts MDM2-p53 interaction, Act.D or H₂O₂-induced p53 accumulation depends on NPM1 translocation.**

9. The authors have not investigated the activation of any of the known p53 target genes such as p21

or PUMA. We therefore do not know how the activity of p53 has changed in relation to the altered levels of p53 protein.

---Following this suggestion, we, based on the experiments with enhanced knockdown and adding-back efficiencies, performed real-time PCR for measurement of mRNA levels of p21 and PUMA. In general, Act.D induced upregulation of p53 targets was blocked in cells bearing C275S NPM1. The data were added as Figure 6e. The mRNA levels p21 and PUMA in the presences of Nutlin-3 plus NAC or DTT were shown as Figure for reviewer 1 i.

10. *Discussion: The authors argue that NPM1 binding to rDNA or rRNA, disrupted by RNase/DNase treatment, that is of critical importance in the translocation of NPM1 to the nucleoplasm. Another possibility that should be mentioned is that RNase treatment destroys pre-ribosomal particles, and this indirectly may cause NPM1 diffusion to the nucleoplasm. The authors hint at this, but I think it should be stated more explicitly.*

---We appreciate this instructive suggestion. We added statements in Results and Discussion.

11. *Methods: The authors need to better describe in the material and methods (Supplementary part), exactly how the RFI (relative fluorescence intensity) was measured and calculated. This is a very central aspect of the paper since many of the images suggest a very strong nucleolar signal.*

---We added the line plot measurement to show the signal intensity in individual cells and modified the description about RFI in Methods (section of Image processing and fluorescence signal quantifications).

12. *The grammar (style) and spelling in the manuscript does not reach up to the high standard one would expect in a Nature Communication publication. The authors need to arrange for editing of the text, as some sections are a bit hard to understand at a first reading.*

---We apologize for non-standard writing. The primary manuscript had been modified by a professional language editing, but further complements might produce some improper sentences. We sent the final version for language editing.

13. *An interesting nonpeptide compound is avrainvillamide, an alkaloid, that targets residue C275 in human NPM1 and also induces p53 activity in cells, this paper published in 2007 (Wulff JE, et al JACS 2007) is included in the references list but the authors may want to discuss their results in relation to the study from 2007 even more. Such discussion is important since it concerns the same residue C275. In my opinion more important than the lengthy discussion on AML and NPMC+.*

---We added discussion on their findings in relation to our results, and shortened the statement about NPMC+.

Reviewer #2 (Redox/ glutathionylation expert):

There is a burgeoning literature detailing how S-glutathionylation of proteins can alter both their structure and function and that the subsequent changes underlie changes in a number of critical cellular pathways. The present communication adds to this by defining how this post-translational modification serves as a general transmitter of stress through the nucleolus. Even though the reports are not discussed in the present paper, there is precedent that S-glutathionylation of key

proteins can impact stress response pathways in other organelles (such as endoplasmic reticulum or mitochondria) and consequently, there is reason to ascribe significance to this post-translational modification in the context of cell stress pathways. In general, the types of stresses used in the present report are quite broad and there is a need to understand how they might coalesce through redox changes that impact specific cysteine residues. In the same context, under the conditions used, how NPM1 glutathionylation might "compete" with similar modifications of other proteins, and the precise mechanism by which cysteine 275 is activated would be informative. In particular, since the authors proceed to study the impact of NPM1 on p53, the fact that the latter is also subject to this post-translational modification might be important to recognize and discuss.

---The reviewer is right that there is precedent that S-glutathionylation of key proteins in endoplasmic reticulum or mitochondria can impact stress response pathways. Our manuscript described S-glutathionylation of a nucleolar protein, probably for the first time. Given that nucleolar stress is a new type of stress but can be induced by various stimuli, our findings that this type of stress can be signified as an oxidative stress in the nucleolus might be interesting. We appreciated the precedent reports on protein S-glutathionylation and now added citation of some of them. However we are not confident that we could figure out how NPM1 glutathionylation might "compete" with similar modifications of other proteins. To partially address this point, we performed IP to determine whether p53 glutathionylation took place under the context, as this reviewer suggested. The results turned out that p53 glutathionylation was detectable before H₂O₂ treatment and remained unchanged after treatment, indicating irrelevance of this p53 modification in the context. Being afraid of that the statement of this result might dilute our emphasis, we didn't put the data to Results; instead, it was presented as **Figure for reviewer 2 a.**

Issues that require some specific attention include:

1. Glutaredoxin has been variously ascribed function(s) in both the forward and reverse reaction of S-glutathionylation. In the case of the glutathionylation cycle, there are enzymes that specifically catalyze the forward reaction. As such, glutaredoxin represents a curious choice in the context of the present study.

---Thanks to the reviewer, these instructive comments helped us to better understand our results of Grx1. We followed this suggestion to interfere an enzyme GSTP that specifically catalyzes forward reaction in glutathionylation cycle. Knockdown of GSTP using two siRNA reached to different silencing efficiencies. A robust decrease of GSTP almost completely abolished H₂O₂-induced NPM1 translocation and p53 accumulation (see **Figure 4c and Supplementary Figure 6b). These results further confirmed the type specificity of oxidative modifications.**

2. A better understanding of how the various types of stress employed in the present series of experiments funnel through redox pathways might be informative. For example, actinomycin D is not generally associated with direct alterations in pathways that would be expected to lead to oxidation of cysteine residues to states that would encourage S-glutathionylation (e.g. creation of cysteinyl residues).

---We agree with these comments. The mechanism underlying the facts that the various types

of stress employed in the present series of experiments funnel through redox pathways must be informative, yet complex and beyond the focus of this study as well. Due to the limitations in the manuscript length and our inputs, it is not feasible to address this question. The editor also suggested not to include this point in the present manuscript. We'll try to get the answer in the future.

3. *Are the authors considering that nucleolar specific pools of GSH/GSSG are critical to the modification of proteins? Does deglutathionylation of NPM1 occur in the nucleolus and result in its translocation back to the cytosol?*

---We think that pools of GSH/GSSG are critical to the NPM1 modification. We applied a fluorescent probe for GSH, but it failed to visualize subcellular differences under both the basal and stress conditions. Please see **Supplementary Figure 4d**. Based on observation that DTT could reverse the stress-induced NPM1 translocation (Fig. 2e, f), we believe that deglutathionylation may mediate recovery of the nucleolar location of NPM1, but we lack approach to dissect where this reaction takes place.

4. *NAC and DTT have some degree of redox specificity for influencing GSH pools and S-glutathionylation. There may be some value in discussing their relative impact on cysteine oxidation and GSH flux.*

---We thank these suggestions and added discussion.

5. *p53 is also S-glutathionylated (Velu et al, Biochemistry, 2007). Is this relevant to the various protein:protein interactions studied here? Also, what is the stoichiometry of NPM1 with p53 and the other proteins in the complex? Is there any competitive process that might influence S-glutathionylation of each protein?*

---We did examine S-glutathionylation of p53 and it remained unchanged under our conditions (**Figure for reviewer 2 a**). Thus we did not go further.

6. *Translocation of NPM1 can also be regulated by JNK. There is literature concerning the importance of redox regulation of JNK and its impact on protein binding partners. It seems possible that such considerations may also be important here?*

---We examined JNK phosphorylation upon H₂O₂ treatment in HeLa and U2OS cells. JNK phosphorylation indeed happened upon this stress and was prevented by its inhibitor SP600125, however, SP600125 couldn't prevent H₂O₂-induced NPM1 translocation. Act.D treatment couldn't even lead to JNK phosphorylation. Please see **Figures for reviewer 2 b-e**. Therefore, JNK is excluded as an upstream regulator in these contexts.

Reviewer #3 (stress and p53 expert):

The ms "Nucleolar stress sensing by NPM1 " address the causes of translocation of nucleolar protein NPM1 to the nucleoplasm that have been described in the literature to occur in response to many types of cellular stresses. However, it is unknown what activates the translocation. These

questions are relevant given that NPM1/NPM1 is a protein that regulates the stabilization and activation of p53, and is also mutated in hematologic cancers. In this work, it is demonstrated that many of the nucleolar stresses alter the redox state of the cells, that these changes occur in the nucleolus, and that NPM1 localization changes are affected by the changes in the redox state. The authors identify that NPM1 C275 undergoes glutathionylation, and show that mutation of this site abrogates NPM1 nucleoplasmic translocation. They demonstrate, using surface plasmon resonance, that the binding of wt NPM1, but not C275S mutant is abrogated by oxidized glutathione (GSSG) in assays using both DNA and RNA oligomers. These findings further support the interpretation that NPM1 nucleolar retention is dependent on its nucleic acid interactions in a manner dependent on its oxidative state. They also address events that lead to p53 stabilization by NPM1, and demonstrate that the C275S mutant does not bind HDM2.

The study is well conceived and conducted and provides a convincing argument supporting NPM1 redox regulation. As such, it provides a significant advance in the field of nucleolar biology.

---We are grateful to the reviewer's positive comments.

Reviewer #4 (Protein-RNA interactions/surface plasmon resonance expert):

Review of manuscript:

Nucleolar stress sensing by NPM1: a redox mechanism underlying its nucleoplasmic translocation and p53 modulation

A. Summary of the key results:

Nucleophosmin (NPM1) is a nucleolar protein involved in cellular stress response and stabilization of p53. During cellular stress NPM1 translocates from the nucleolus to the nucleoplasm and this is key to p53 stabilization. Yet how this translocation is mediated is mechanistically unclear and is the focus of this manuscript. In this manuscript the authors examine the cause of translocation of NPM1 from the nucleolus to the nucleoplasm. The authors use innovative colorimetric single cell assays and a variety of tests and mutants to provide data in support of the idea that NPM1 translocation is controlled by a redox mechanism involving cysteine 275, and that this phenomenon could play a role in stabilization of p53.

B. Originality and interest:

The results are interesting, potentially important and to the knowledge of this reviewer, novel.

---We are grateful to the reviewer's positive comments.

C. Data & methodology: validity of approach, quality of data, quality of presentation:

1. The authors' assertion that NPM1 becomes dispersed in the nucleus upon stress appears to be true for hydrogen peroxide but not that convincingly for actinomycin D.

---We admitted that 500 μ M H₂O₂ was a potent inducer for NPM1 translocation. Yet actinomycin D could similarly cause the phenomenon, but to a milder extent. This was also told by statistical analysis in Figure 2a. The images in Figure 2a for Act.D were not representative enough. We now re-selected the images, and presented the data with line plot showing the signal intensities in the nucleoli and nucleoplasm respectively (Supplementary Figure 2a, 6a).

2. The experiments in Figure 2 were carried out with hydrogen peroxide, again supporting the idea that this is the stressor that should be used in all experiments.

--- Figure 6 was originally organized to mimic the most typical and frequently-used nucleolar stress condition created by Act.D. We followed this reviewer's suggestion to perform some in parallel experiments using H₂O₂ (Figure 6b, d, Supplementary figure 6d and Figure for reviewer 5 b).

3. Given the importance of technique in the execution and interpretation of SPR experiments, the authors should provide more details in the methods section accompanying the actual manuscript, as opposed to in the supplementary information.

--We modified the description. But due to strict word number limitation, we could not provide more details in the actual manuscript.

The mathematical model of the SPR experiment overlaid on the curves does not fit the data well. The modeled dissociation goes to zero whereas this does not happen in the actual data. This issue should be addressed by the authors. From the online methods section it is unclear how the authors calculated the K_D , since it is stated that two methods were used. The first method (using association and dissociation rates) would not provide accurate values given the fact that the modeled dissociation rates do not fit the data. The surface density coated by the investigators is too high which may affect the data quality. This is reinforced by the fact that lower concentrations of protein appear to give a better fit. A much lower surface density would still provide high enough signal to give curves that can be interpreted. It is thus recommended that the investigators coat tenfold less DNA or RNA on their sensor chips.

---We admire this reviewer's expertise in this technique. We know that the quality of our SPR experiments was not satisfactory; we were not skillful in manipulation and optimization. Here we wanted to clarify two points. First, due to the aims of the assay, we experienced difficulties in the purity and yield of the protein samples. The routine GST method for purification needs to add GSH that may interfere the GSSG effect in our experimental settings. Trying to eliminate GST or GSH contamination in the samples, we conducted a thorough wash and thus got very low quantity that won't allow titration assays. Eventually, our proteins were constructed with SUMO tag and purified at a compromised purity for some unknown reason. We had to increase a bit the concentrations of rDNA and rRNA on the sensor chips. This made the signal/noise ration unsatisfying, there might be unspecific binding with the chips, but we had difficulties to optimize. Under this condition, we could not get the usual lower K_{on} and ideal zero values of dissociation. Second, the mathematical model in the previous sensorgrams was incorrectly used; the curves were fit to real association/dissociation trends in our assays, but the dissociation ratios were set to zero, thus these "standard" curves did not fit the data well. The K_D values were not calculated by these "standard" curves, instead, by one-site steady state modeling.

Therefore, this time we presented the sensorgrams of the actual values (Fig. 5g), and showed the plots that were used for K_D calculation with the one-site steady state modeling, $Y=B_{max} * X / (K_D + X) + background$ (Supplementary figure 5e). We also rewrote the descriptions of the calculation for the K_D in Supplementary information and moved the description of SPR to the Methods in text.

We hope that the reviewer would agree that this data, despite of less accurate, didn't impair the justification of the conclusions.

4. The authors transition to the use of actinomycin D in the end of the manuscript, at Figure 6. If stress causes p53 stabilization, then this should be observed with H₂O₂ treatment as well. All previous experiments were carried out with peroxide. Actinomycin D is highly toxic, affects the production of all proteins through its inhibition of RNA polymerase, and thus results obtained with actinomycin D can be artifactual. All experiments should at least be carried out with the same compound (H₂O₂).

--- As we replied above, Figure 6 was originally designed to mimic the most typical nucleolar stress condition created by Act.D. The doses used were low, free of the effects of apoptosis, and believed to selectively suppress Polymerase I (Perry and Kelley, 1970). We followed this reviewer's suggestion to perform some in parallel experiments using H₂O₂ (Figure 6b, d, Supplementary figure 6d and Figure for reviewer 5 b). However, doubling all experiments may produce a too large body of data, especially images, which seems awkward.

5. The authors refer to the use of Actinomycin D in clinical trials. How does the dose used in their experiments compare with the clinically used dose?

---The dose used in the reported clinical trials was approximately 150 ng/ml (12.5 µg/kg × 60 kg / 5000 ml), while our in vitro dose was 10 ng/ml.

D. Appropriate use of statistics and treatment of uncertainties:

6. The authors should clarify the statistical significance designation in Figure 1. For example, they should indicate whether any of the NAC-treated cells are significantly different from control NAC-treated cells. It is noted that in the presence of UV, the treated sample appears to be higher than the control sample.

---We added these statistics in Figure legend of Figure 1b and described the trends in the text. The nuclei appeared more oxidation prone in the presences of H₂O₂, UV and heat, and the reversal by NAC were less efficient in the groups of UV and heat. The values higher than the control indicated the partial reversal, which was also true for the some other groups.

7. In the last sentence on page 8 the authors state that the modification "apparently impeded the association of the interface with the potential nuclear macromolecules", however the data shown is a simulation and should only be stated to suggest a mechanism.

---We modified the statements.

E. Conclusions: robustness, validity, reliability:

8. In the first full paragraph of page 6 of the manuscript the authors suggest that the observations in the presence of DTT have a strong causal implication, however these observations are merely correlations. It is suggested that this sentence be reworded.

---We modified the statements.

9. On page 9 the authors imply their observations of cells treated with RNAs and DNAs to show a causal relationship when in fact this is an association under conditions that are highly abnormal. The text should be more carefully worded using the words "suggest" rather than "confirmed".

---We modified the statements.

10. *At the top of page 10, beginning of the paragraph, the authors should use the word "test" as opposed to "verify".*

---We corrected.

F. Suggested improvements: experiments, data for possible revision:

Please see all issues mentioned above.

In addition:

11. *The authors should use the HUGO nomenclature NPM1 for the gene nucleophosmin.*

---We replaced B23 with NPM1 all through the manuscript.

12. *The use of the English language by the authors could use some polishing and reviewed by a native speaker is recommended.*

--We sent the manuscript to a language editing.

13. *The authors are encouraged to use different color schemes for different treatments in Figure 5C. The use of similar colors for different treatments in the adjacent panels is confusing.*

--We changed the colors.

G. References: appropriate credit to previous work?

The authors appear to reference previous work correctly.

H. Clarity and context:

14. *The abstract requires clarification in that it is unclear that the authors are referring to a different mutant (than the Cys275 one) when they discuss "the mutant protein with nucleoplasmic location".*

--We modified.

15. *In the discussion, the following point should be clearly addressed: How does the dose Actinomycin D dose used in their experiments compare with the clinically used dose?*

--We added description.

Reviewer #5 (stress and p53 expert):

The manuscript entitled "Nucleolar stress sensing by NPM1: a redox mechanism underlying its nucleoplasmic translocation and p53 modulation" describes investigations in the role of the protein NPM1 in sensing nucleolar mediated stress response upon different stimuli. The Authors have used different stimuli (H₂O₂, hypoxia, UV irradiation, heat shock, starvation and Act. D) to change the cell redox state. They show that the change in redox state causes the release of NPM1 from the nucleolus and that this release requires a post-translational modification of cysteine 275 which, upon stress, becomes S-glutathionylated. Furthermore, the Authors show that the release of NPM1 from the nucleolus consistently coincides with loss of rDNA and rRNA binding. Thus, the Authors

postulate that the already known ability of NPM1 to bind nucleic acids is at the basis of its location in the nucleolus, and that its relocation to the cytoplasm is due to S-glutathionylation of cysteine 275 upon stress, which, in turn, causes its dissociation from rRNA and rDNA, as shown by mutagenesis experiments. Finally, the Authors address the effect on p53 stabilization of the translocation of NPM1 to the cytoplasm, since NPM1 translocation and the activation of p53 as a stress response have been often described to occur together under nucleolar stress. They conclude that a stress mediated NPM1 nucleolar release is required for the regulation of p53 activation under stress.

The manuscript is clearly and logically presented. However, while the Authors have provided convincing and detailed data on the mechanism underlying the release of NPM1 from the nucleolus upon several stress stimuli, as well as on the role of S-glutathionylation in mediating the interaction of NPM1 with nucleic acids, the data on the ability of NPM1 to bind rDNA and rRNA or to interfere with the p53-HDM2 interaction upon stress dependent release, with consequent p53 accumulation, are not novel (Kurki S. et al, Cancer Cell, 2004, 5: 465-75, also cited by the Authors). Notably, the Authors show that nucleoplasmic localization of NPM1 is necessary but not enough to induce p53 accumulation, indicating that others stress induced mechanisms might be involved.

A limit of this manuscript is indeed the lack of real investigations in the mechanisms of p53 stabilization by NPM1.

--It is true that the ability of NPM1 to interfere with the p53-HDM2 interaction has been delicately demonstrated in early time. NPM1 can, via its C-terminal domain, interact with p53 (Colombo E, et al., 2002). NPM1 is also found to bind with HDM2, competing with p53 and blocking HDM2-mediated ubiquitination of p53 (Kurki et al., 2004). Let alone the controversy of these conclusions, these author didn't address how a nucleolar protein might act its nucleoplasmic function under stresses by interfering p53-HDM2 interaction. They even ignored the subcellular compartments that these interactions might take place. Our claims that NPM1 translocation is a release from the binding with rDNA/rRNA had not been proposed, and that the release is following oxidation had never been assumed either. When other authors claimed that the nucleoplasmic presence of the ribosomal proteins is required for p53 accumulation under the nucleolar stress (Jin et al., 2004), the role of NPM1 became ambiguous. So, the emphasis of the last part of this manuscript is to connect the NPM1 translocation, following its oxidation, with the interruption of p53-HDM2. We strengthened this part regarding the mechanisms of p53 stabilization by NPM1 in the revised version. We compared the way of NPM1 interact with p53-HDM2 with Nutilin-3, a reagent supposed to directly disrupt p53-HDM2 interaction, and we also, following this reviewer's suggestion, put another regulator, ARE, into consideration.

As for the comments that "nucleoplasmic localization of NPM1 is necessary but not enough to induce p53 accumulation, indicating that others stress induced mechanisms might be involved", we would argue as below. The notion that a simple nucleoplasmic localization of NPM1 is insufficient to induce p53 accumulation was hinted in our study, based on the use of the W288/290A mutant. This mutant couldn't cause p53 accumulation unless in the presence of the nucleolar stress stimuli. We postulate that the stress stimuli may simultaneously exert some additional modification(s) to p53 when they induce NPM1 translocation. Under a real nucleolar stress condition, these two pathways inevitably act together, thus this notion does not conflict or weaken the importance of our findings about necessity of the NPM1 translocation.

In particular:

1. - In Fig. 6, in order to correlate NPM1 localization, as well as that of the mutants, with p53 stabilization, the Authors should perform cellular fractionation to separate the nucleus from the cytoplasm and show their levels in the two compartments.

--We performed this experiment. The results showed that Act.D induced p53 accumulation in the nucleus (**Figure for reviewer 5 a**).

2- The Authors should verify whether p53 is only phosphorylated or acetylated (posttranslational modification by which p53 stabilization and function are regulated) in the presence of NPM1

--We performed this experiment. The results showed that H₂O₂ and Act.D induced p53 accumulation accompanied by increased p53 phosphorylation and acetylation; in the cells transfected with C275S mutants, the accumulation of p53 was blocked, and two modifications were simultaneously less visible (**Figure for reviewer 5 b**). We don't want to conclude that NPM1 translocation regulates these modification; the changes seem due to the amounts of p53 proteins.

3- The Authors should also verify p53 levels of activity evaluating the expression of its transcriptional targets.

--We performed this experiment. The results of real-time PCR were presented as **Figures 6e**.

-4. The Authors should assess the role of ARF (which has a key function in the p53-mdm2-NPM1 pathway) by using a cell line expressing it, since U2OS are ARF negative.

--There have been impressive publications reporting the role of ARF in the p53-MDM2-NPM1 interactions. But, it is a big challenge to find proper cell types to clarify their reciprocal regulation; there is few cell types bearing both nucleolar ARF and the wild type p53. During the study we couldn't see the nucleolar localization of ARF in HeLa cells; ARF was intensely stained only in the nucleoplasm (**Figure for reviewer 5 c**), which differed from the claims of previous investigators (**Lindstrom et al., 2000**). As these and other authors used human and mouse fibroblasts to investigate ARF, in response to this reviewer's requirement we determined p53 levels in an immortalized MEF cell line expressing ARFs. We found the patterns that Act.D induced p53 accumulation remained similar to those in U2OS cells free of ARF (**Figure for reviewer 5 d**). We had asked for cell lines and antibodies from the colleagues and test some available cells, which gave no clear conclusion and made us to postpone the date of resubmission. So we felt difficulty to go further in the interrelationship of ARF with NPM1 and p53 in limited time.

- The Authors should assess the effect of other p53 stabilizing treatments utilizing alternative oxidative stimuli (such as H₂O₂), which also induce S-glutathionylation of NPM1, in order to confirm the act. D data.

--We performed the in parallel experiment using H₂O₂, and the results were presented as **Figure 6b, d, Supplementary figure 6d and Figure for reviewer 5 b**.

They should also examine whether this pathway is active only in the presence of oxidative stress, by

using different stimuli which do not produce oxidative stress, S-glutathionylation of NPM1, and/or translocation of NPM1 from the nucleolus, but can stabilize p53, likely through other pathways.

--This reviewer's suggestion coincided with another reviewer who asked for use of Nutlin-3. We took Nutlin-3 that had been previously proven disrupted MDM2-p53 interaction directly (Anil et al., 2013). With this control, we can conclude that, differently with Nutlin-3 that directly disrupts MDM2-p53 interaction, Act.D or H₂O₂-induced p53 accumulation depends on NPM1 translocation. See **Figure for reviewer 1 f-h and Figure 6d**.

Minor points

5- Fig. S5: The Authors use GADPH as negative control of RIP and ChIP experiments, however, we do not think this is appropriate as rDNA is present in hundreds of copies in the genome. As negative control, the Authors should use a region of rDNA or rRNA where they do not expect to find NPM1 binding.

--We noticed during the ChIP work that NPM1 had a wild but somewhere weaker binding with the full-length rDNA genes; apart from the stronger binding at the specific elements. The data were presented for this reviewer as **Figure for reviewer 5 e**. This was the reason we didn't choose a rDNA element as the negative control.

- Anil, B., C. Riedinger, J.A. Endicott, and M.E. Noble. 2013. The structure of an MDM2-Nutlin-3a complex solved by the use of a validated MDM2 surface-entropy reduction mutant. *Acta crystallographica. Section D, Biological crystallography*. 69:1358-1366.
- Dooley, C.T., T.M. Dore, G.T. Hanson, W.C. Jackson, S.J. Remington, and R.Y. Tsien. 2004. Imaging dynamic redox changes in mammalian cells with green fluorescent protein indicators. *The Journal of biological chemistry*. 279:22284-22293.
- Jin, A., K. Itahana, K. O'Keefe, and Y. Zhang. 2004. Inhibition of HDM2 and activation of p53 by ribosomal protein L23. *Molecular and cellular biology*. 24:7669-7680.
- Kurki, S., K. Peltonen, L. Latonen, T.M. Kiviharju, P.M. Ojala, D. Meek, and M. Laiho. 2004. Nucleolar protein NPM interacts with HDM2 and protects tumor suppressor protein p53 from HDM2-mediated degradation. *Cancer cell*. 5:465-475.
- Lindstrom, M.S., U. Klangby, R. Inoue, P. Pisa, K.G. Wiman, and C.E. Asker. 2000. Immunolocalization of human p14(ARF) to the granular component of the interphase nucleolus. *Experimental cell research*. 256:400-410.
- Perry, R.P., and D.E. Kelley. 1970. Inhibition of RNA synthesis by actinomycin D: characteristic dose-response of different RNA species. *Journal of cellular physiology*. 76:127-139.

Figures for reviewer 1 | **(a)** NLS-roGFP1 was expressed in HeLa cells, then nucleolar (blue) and nucleoplasmic (red) redox states were calculated and plotted against each individual cell. n=17. **(b)** After additional H₂O₂ (500 μM) treatment, relative nucleolar and nucleoplasmic redox changes in 20 minutes were showed as line plot. **(c)** Heterologomerization of NPM1 WT and C275S mutant. Different tags of NPM1 WT and C275S mutant were transfected to HEK293T cells and GFP-NPM1 WT was immunoprecipitated by anti-FLAG M2 beads. **(d)** Heterologomerization of S-glutathionylated GFP-NPM1 and FLAG-NPM1, detected by non-reduced immunoprecipitation with anti-FLAG antibody. **(e)** S-glutathionylation state changes of FLAG-NPM1 WT and C275S mutant after H₂O₂ (500 μM) treatment ± NAC (5mM) pretreatment, detected by non-reduced immunoprecipitation with anti-FLAG antibody. FLAG-NPM1 WT and C275S mutant were transfected to NPM1-silenced U2OS cell line (shNPM1 (5'UTR)) and S-glutathionylated FLAG-NPM1 were detected with anti-glutathione antibody. INPUT samples was mixed with reduced loading buffer. **(f)** ROS level changes upon 1 or 4 h Nutlin-3 (20 μM) treatment, detected with DCFH-DA from three independent experiments. Unpaired *t* test, *P* value was showed above the bars. **(g)** Endogenous NPM1 localization and nucleolar line profiles of NPM1 in representative cells upon 4 h Nutlin-3 treatment (left). The RFI of the nucleolar NPM1 was displayed (right). n = 57. Unpaired *t* test, *P* value was showed above the bars. **(h)** p53 accumulation upon Nutlin-3 treatment. Cells were pretreated with NAC (5 mM) 4 h prior to Nutlin-3 treatment, or co-treated with DTT (5 mM) and Nutlin-3. **(i)** Relative mRNA level changes of p21 and PUMA after Act.D (8 nM) or Nutlin-3 (20 μM) treatments ± NAC (5 mM) pretreatment or ± DTT (5 mM) treatment. Unpaired *t* test, *P* < 0.01 (**) or < 0.001 (***) or showed above bars with respected to untreated vs. treated groups. Data are represented as mean ± s.e.m.

Figures for reviewer 2 | (a) S-glutathionylation state of endogenous NPM1 and p53 after H₂O₂ (500 μM) treatment, detected by non-reduced immunoprecipitation with anti-glutathione antibody. (b) JNK phosphorylation states after H₂O₂ (500 μM) treatment ± JNK inhibitor SP600125 (10 μM) in 30 mins. (c) Endogenous NPM1 localization and nucleolar line profiles of NPM1 in representative cells after H₂O₂ (500 μM) treatment ± SP600125. (d) The RFI of the nucleolar NPM1 in HeLa and U2OS cells after H₂O₂ (500 μM) treatment ± SP600125 were displayed. n > 55. Unpaired *t* test, *P* < 0.001 (***) compared to control group. Data are represented as mean ± s. e. m. (e) JNK phosphorylation was not changed after Act.D (8 nM) treatment, while p53 accumulation was detectable. SP, SP600125 (#sc-200635, Selleck, USA).

Figures for reviewer 5 | **(a)** Nuclear and cytosol fractions of U2OS cell treated with Act.D (8 nM) with or without NAC pretreatment or DTT co-treatment. Antibody PARP (#9532) was obtained from CST. **(b)** Accumulation, acetylation (K382) and phosphorylation (S15) of p53 in NPM1-silenced U2OS cells (shNPM1 5'UTR) that added back FLAG-NPM1 WT or C275S mutants before treatment with Act.D (8 nM, upper), H₂O₂ (500 μ M, bottom) or Nutlin-3 (20 μ M). Antibody Phospho-p53(S15) (#9284) and Acetyl-p53(K382) (#2525) were all obtained from CST. **(c)** Localization of p14/ARF in HeLa cells \pm H₂O₂ (500 μ M) treatment, stained with anti-p14/ARF (#2407, CST) antibody. **(d)** Endogenous p53 accumulation in immortalized MEF cells that transiently transfected with FLAG-NPM1 WT or C275S mutants before treatment with Act.D (8 nM), blotted with anti-p53 (#2524, CST) antibody. **(e)** The relative quantities of rDNA bound with FLAG-NPM1 in HEK293T were assessed. Up to 16 primers (according to reference, PMID: 18332108) were used to cover the whole 47S genome, including the coding region. Data are represented as mean \pm s.e.m.

Reviewers' Comments:

Reviewer #1 (Remarks to the Author)

(Reviewer 1)

I find that the authors have addressed my concerns in an appropriate manner. The paper contains several interesting findings and is a timely, original, contribution to the research field. Data presentation is clear and of excellent quality.

I have a few minor suggestions and comments that the authors may want to incorporate:

1)Evidently NPM1 translocation is linked to the "status" of the C275 residue, but it is a bit troublesome that the C275S mutant did not result in a loss of NPM1 glutathionylation (as shown in the figure for reviewer R1). One can envisage that the other potential sites are indeed modified or that compensatory modification occurs. I would encourage the authors to mention this result in the text and show the co-IP data in the supplement (not only for reviewer). I do not however consider it strictly necessary since this issue has now been discussed in this rebuttal letter - communication. The authors should have tested the 3C/S mutant to see whether glutathionylation was lost completely.

2)Recently it was reported that oxidative stress triggers phospho-modification of NPM1 Thr199 residue. While that study does not have any major implications for this study, it may be good to mention it (Guillonneau M, et al, FASEB J, 2016, in press).

3)The authors use the word "universal" in the abstract and figure 1 which I consider a bit too strong. Maybe a better word would be "common", or "general", in fact most experiments in the paper were done using H₂O₂ and/ or Act D as agents. Also, I wonder if the word in the abstract "retains" is appropriate style (remains?). Please consider re-writing that sentence.

4)Please also add a few sentences to the figure 7 model legend, describing the model. Now there is only a title. It would serve as a brief summary.

Reviewer #2 (Remarks to the Author)

This version of the paper is more complete and continues to address an interesting question related to cellular stress response. The primary unanswered question remains how such a diverse range of chemical and nutritional insults might coalesce into modifying oxidative conditions within the cell (and presumably specifically the nucleolus). Obviously, hydrogen peroxide will diffuse and activate various cysteine residues (including cys 275 of NPM1) forming the thiolate anion, which will be a prerequisite for S-glutathionylation. Quite how other divergent stressers achieve this is not really addressed. However, to explain fully how this oxidative modification is achieved, thus providing a centralized general stress response is a critical question.

Other issues:

Line 49: It is worth noting that S-glutathionylation of NPM1 was initially reported in 2006 following exposure alkylating agent stress (Molecular Pharmacology: 69: 501-8, 2006). Indeed, this adds a further electrophilic chemical stress to the list

P53 is also subject to S-glutathionylation and it is likely worth discussing whether this has any relevance in the present circumstances.

Line 174 etc: There are more than 10 other PTM of cysteines that might occur. Moreover, cys275 is not a single residue, necessarily excluding it from forming disulfides with the other cys in NPM or with other protein cys through cross-links. The discussion of NPM1 oligomeric structures should include this.

Grx has been reported to catalyze both glutathionylation and deglutathionylation. In the context of its use in the present experiments, some discussion of this would be useful. There are numerous grammatical and narrative errors likely resulting from the authors having English as a second language.

Reviewer #4 (Remarks to the Author)

Re-Review NPM1 manuscript

Title: A redox mechanism underlying the nucleolar stress sensing by nucleophosmin.

This is a re-review of a previously submitted manuscript. The authors have responded well to the reviewers' criticism, and have provided additional data in support of their findings.

A. Summary of the key results:

Nucleophosmin (NPM1) is a nucleolar protein involved in cellular stress response and stabilization of p53. During cellular stress NPM1 translocates from the nucleolus to the nucleoplasm and this is key to p53 stabilization. Yet how this translocation is mediated is mechanistically unclear and is the focus of this manuscript. In this manuscript the authors examine the cause of translocation of NPM1 from the nucleolus to the nucleoplasm. The authors use innovative colorimetric single cell assays and a variety of tests and mutants to provide data in support of the idea that NPM1 translocation is controlled by a redox mechanism involving cysteine 275, and that this phenomenon could play a role in stabilization of p53.

B. Originality and interest:

The results are interesting, potentially important and to the knowledge of this reviewer, novel.

C. Data & methodology: validity of approach, quality of data, quality of presentation:

Critique: The authors' assertion that NPM1 becomes dispersed in the nucleus upon stress appears to be true for hydrogen peroxide but not that convincingly for actinomycin D. Response: The authors have addressed this point by selecting more representative images.

Critique: The experiments in Figure 2 were carried out with hydrogen peroxide, again supporting the idea that this is the stressor that should be used in all experiments.

Response: The authors have addressed this point with added data.

Critique: Given the importance of technique in the execution and interpretation of SPR experiments, the authors should provide more details in the methods section accompanying the actual manuscript, as opposed to in the supplementary information. The mathematical model of the SPR experiment overlaid on the curves does not fit the data well. The modeled dissociation goes to zero whereas this does not happen in the actual data. This issue should be addressed by the authors. From the online methods section it is unclear how the authors calculated the KD, since it is stated that two methods were used. The first method (using association and dissociation rates) would not provide accurate values given the fact that the modeled dissociation rates do not fit the data. The surface density coated by the investigators is too high which may affect the data quality. This is reinforced by the fact that lower concentrations of protein appear to give a better fit. A much lower surface density would still provide high enough signal to give curves that can be interpreted. It is thus recommended that the investigators coat tenfold less DNA or RNA on their sensor chips.

Response: The authors have addressed this point by adding a more detailed description of the methods, and they acknowledge problems fitting the data so that they limit their interpretation. This is a valid approach, as their curve fitting was inappropriate. The current data presentation is acceptable.

Critique: The authors transition to the use of actinomycin D in the end of the manuscript, at Figure 6. If stress causes p53 stabilization, then this should be observed with H₂O₂ treatment as well. All previous experiments were carried out with peroxide. Actinomycin D is highly toxic, affects the production of all proteins through its inhibition of RNA polymerase, and thus results obtained with

actinomycin D can be artifactual. All experiments should at least be carried out with the same compound (H₂O₂).

Response: the authors have added additional data with H₂O₂.

Critique: The authors refer to the use of Actinomycin D in clinical trials. How does the dose used in their experiments compare with the clinically used dose?

Response: The authors answered this question.

D. Appropriate use of statistics and treatment of uncertainties:

Critiques: The authors should clarify the statistical significance designation in Figure 1. For example, they should indicate whether any of the NAC-treated cells are significantly different from control NAC-treated cells. It is noted that in the presence of UV, the treated sample appears to be higher than the control sample. In the last sentence on page 8 the authors state that the modification "apparently impeded the association of the interface with the potential nuclear macromolecules", however the data shown is a simulation and should only be stated to suggest a mechanism.

Response: Authors followed the reviewer's suggestions.

E. Conclusions: robustness, validity, reliability

Critiques: In the first full paragraph of page 6 of the manuscript the authors suggest that the observations in the presence of DTT have a strong causal implication, however these observations are merely correlations. It is suggested that this sentence be reworded. On page 9 the authors imply their observations of cells treated with RNAs and DNAs to show a causal relationship when in fact this is an association under conditions that are highly abnormal. The text should be more carefully worded using the words "suggest" rather than "confirmed". At the top of page 10, beginning of the paragraph, the authors should use the word "test" as opposed to "verify".

Response: Authors followed the reviewer's suggestions.

F. Suggested improvements: experiments, data for possible revision:

Please see all issues mentioned above. In addition:

Critiques: The authors should use the HUGO nomenclature NPM1 for the gene nucleophosmin. The use of the English language by the authors could use some polishing and reviewed by a native speaker is recommended. The authors are encouraged to use different color schemes for different treatments in Figure 5C. The use of similar colors for different treatments is confusing.

Response: Authors followed the reviewer's suggestions.

G. References: appropriate credit to previous work?

The authors appear to reference previous work correctly.

H. Clarity and context:

Critiques: The abstract requires clarification in that it is unclear that the authors are referring to a different mutant (than the Cys275 one) when they discuss "the mutant protein with nucleoplasmic location". In the discussion, the following point should be clearly addressed: How does the dose Actinomycin D dose used in their experiments compare with the clinically used dose?

Response: Authors followed the reviewer's suggestions.

In summary, the authors' responses to the critiques on data and methodology were acceptable.

Clearly, the authors put a strong effort into addressing all the reviewer's criticisms. The manuscript has improved considerably.

Reviewer #5 (Remarks to the Author)

In their revised submission, the Authors addressed several of our concerns; however, we still feel that this work does not add further insights to the already known mechanism of p53 stabilization.

In detail:

1. We had asked the Authors to assess the role of ARF (which has a key function in the p53-

mdm2-NPM1 pathway) by using a cell line expressing it, since U2OS are ARF negative. The Authors state: "During the study we couldn't see the nucleolar localization of ARF in HeLa cells; ARF was intensely stained only in the nucleoplasm" . Since this result is in contrast with published reports, did the Authors consider that this observation might have been due to non-specific staining? We are also surprised that the Authors chose HeLa cells where p53 activity levels are known to be difficult to establish due to inhibition by the HPV oncoprotein E6. The Authors do not specify which cell lines they received from colleagues to try to assess ARF role; were they able to test the U2OS/Tet-On/p19ARF-inducible cell line (doi:10.4161/auto.25831), or the inducible MT-Arf cell line (a NIH-3T3 cell derivative; doi:10.1101/gad.1213904)?

2. Fig.6/Figure for reviewer 5 a: The indicated experiment only shows p53 (not NPM or NPM mutant) thus preventing any correlation between NPM and p53 localization

3. Fig. S5: We had suggested that the Authors could use as negative control a region of rDNA or rRNA where they did not expect to find NPM1 binding. In their rebuttal the Authors state: "We noticed during the ChIP work that NPM1 had a wild but somewhere weaker binding with the full-length rDNA genes; apart from the stronger binding at the specific elements. The data were presented for this reviewer as Figure for reviewer 5 e. This was the reason we didn't choose a rDNA element as the negative control." However, as the 35S ribosomal DNA (rDNA) units, repeated in tandem at one or more chromosomal loci, are separated by an intergenic spacer (IGS) and the IGS is a repeated but not transcribed region, we believe it could have been used as a negative control.

point-by-point response

Reviewer #1 (Remarks to the Author):

(Reviewer 1)

I find that the authors have addressed my concerns in an appropriate manner. The paper contains several interesting findings and is a timely, original, contribution to the research field. Data presentation is clear and of excellent quality.

I have a few minor suggestions and comments that the authors may want to incorporate:

1)Evidently NPM1 translocation is linked to the "status" of the C275 residue, but it is a bit troublesome that the C275S mutant did not result in a loss of NPM1 glutathionylation (as shown in the figure for reviewer R1). One can envisage that the other potential sites are indeed modified or that compensatory modification occurs. I would encourage the authors to mention this result in the text and show the co-IP data in the supplement (not only for reviewer). I do not however consider it strictly necessary since this issue has now been discussed in this rebuttal letter - communication. The authors should have tested the 3C/S mutant to see whether glutathionylation was lost completely.

---We are afraid of that presenting and discussion of this issue would dilute our emphasis that glutathionylation at one cysteine can induce NPM1 translocation. Thus we prefer not to show the data and expand discussion. We also apologize for not performing the co-IP using the 3C/S mutant to see whether glutathionylation was lost completely. Because cysteine residuals are the only sites of glutathionylation, it is predictable that glutathionylation will lost completely in the 3C/S mutant. We hope this reviewer will agree to our consideration.

2)Recently it was reported that oxidative stress triggers phospho-modification of NPM1 Thr199 residue. While that study does not have any major implications for this study, it may be good to mention it (Guillonneau M, et al, FASEB J, 2016, in press).

---We added citation of this fresh document in Instruction.

3)The authors use the word "universal" in the abstract and figure 1 which I consider a bit too strong. Maybe a better word would be "common", or "general", in fact most experiments in the paper were done using H2O2 and/ or Act D as agents. Also, I wonder if the word in the abstract "retains" is appropriate style (remains?). Please consider re-writing that sentence.

---We appreciated these suggestions and changed two words (see revised in red).

4)Please also add a few sentences to the figure 7 model legend, describing the model. Now there is only a title. It would serve as a brief summary.

---We added description as Figure legends for figure 7.

Reviewer #2 (Remarks to the Author):

This version of the paper is more complete and continues to address an interesting question related to cellular stress response. The primary unanswered question remains how such a diverse range of chemical and nutritional insults might coalesce into modifying oxidative conditions within the cell (and presumably specifically the nucleolus). Obviously, hydrogen peroxide will diffuse and activate various cysteine residues (including cys 275 of NPM1) forming the thiolate anion, which will be a prerequisite for S-glutathionylation. Quite how other divergent stressers achieve this is not really addressed. However, to explain fully how this oxidative modification is achieved, thus providing a centralized general stress response is a critical question.

---These comments are instructive; we should try to address the critical question how divergent stressers achieve oxidative modification of proteins in the future.

Other issues:

Line 49: It is worth noting that S-glutathionylation of NPM1 was initially reported in 2006 following exposure alkylating agent stress (Molecular Pharmacology: 69: 501-8, 2006). Indeed, this adds a further electrophilic chemical stress to the list.

---We added citation of this paper and a brief discussion (see page 16).

P53 is also subject to S-glutathionylation and it is likely worth discussing whether this has any relevance in the present circumstances.

--- We agree with the comments that S-glutathionylation of p53 is worth discussing, however, given that S-glutathionylation of p53 remains unchanged under our conditions, this event is irrelevant to our findings. We prefer not to discuss this phenomenon, because the data presentation in the figures will dilute our major findings.

Line 174 etc: There are more than 10 other PTM of cysteines that might occur. Moreover, cys275 is not a single residue, necessarily excluding it from forming disulfides with the other cys in NPM or with other protein cys through cross-links. The discussion of NPM1 oligomeric structures should include this.

---We had excluded the possibility of disulfide formation before narrowed down to glutathionylation during experiments, which was described in Results (see page 8 “Because the responsible cysteine residue was single, the possibility of a formation of an intramolecular disulfide bond was excluded. The oligomerization (to form pentamers) of NPM1 was unaffected by redox changes²⁸; therefore, the formation of intermolecular disulfides between two or more NPM1 was also unlikely.”). Although we had no evidence or assumption to exclude that Cys275-based disulfide might mediate cross-links of NPM1 with other proteins, the final confirmation of S-glutathionylation on this cysteine automatically exclude this possibility. So, it will be logically improper to discuss this point in Discussion; instead, we add a sentence to mention this in Results, as one point to be considered in process of reasoning (see page 8 in red).

Grx has been reported to catalyze both glutathionylation and deglutathionylation. In the context of its use in the present experiments, some discussion of this would be useful.

---We added discussion.

There are numerous grammatical and narrative errors likely resulting from the authors having English as a second language.

---The revised manuscript has been language edited by a professional service company (pls see the certificate).

Reviewer #4 (Remarks to the Author):

Re-Review NPM1 manuscript

Title: A redox mechanism underlying the nucleolar stress sensing by nucleophosmin.

This is a re-review of a previously submitted manuscript. The authors have responded well to the reviewers' criticism, and have provided additional data in support of their findings.

A. Summary of the key results:

Nucleophosmin (NPM1) is a nucleolar protein involved in cellular stress response and stabilization of p53. During cellular stress NPM1 translocates from the nucleolus to the nucleoplasm and this is key to p53 stabilization. Yet how this translocation is mediated is mechanistically unclear and is the focus of this manuscript. In this manuscript the authors examine the cause of translocation of NPM1 from the nucleolus to the nucleoplasm. The authors use innovative colorimetric single cell assays and a variety of tests and mutants to provide data in support of the idea that NPM1 translocation is controlled by a redox mechanism involving cysteine 275, and that this phenomenon could play a role in stabilization of p53.

B. Originality and interest:

The results are interesting, potentially important and to the knowledge of this reviewer, novel.

C. Data & methodology: validity of approach, quality of data, quality of presentation:

Critique: The authors' assertion that NPM1 becomes dispersed in the nucleus upon stress appears to be true for hydrogen peroxide but not that convincingly for actinomycin D. Response: The authors have addressed this point by selecting more representative images.

Critique: The experiments in Figure 2 were carried out with hydrogen peroxide, again supporting the idea that this is the stressor that should be used in all experiments.

Response: The authors have addressed this point with added data.

Critique: Given the importance of technique in the execution and interpretation of SPR experiments, the authors should provide more details in the methods section accompanying the actual manuscript, as opposed to in the supplementary information. The mathematical model of the SPR experiment overlaid on the curves does not fit the data well. The modeled dissociation goes to zero whereas this does not happen in the actual data. This issue should be addressed by the authors. From the online methods

section it is unclear how the authors calculated the K_D , since it is stated that two methods were used. The first method (using association and dissociation rates) would not provide accurate values given the fact that the modeled dissociation rates do not fit the data. The surface density coated by the investigators is too high which may affect the data quality. This is reinforced by the fact that lower concentrations of protein appear to give a better fit. A much lower surface density would still provide high enough signal to give curves that can be interpreted. It is thus recommended that the investigators coat tenfold less DNA or RNA on their sensor chips.

Response: The authors have addressed this point by adding a more detailed description of the methods, and they acknowledge problems fitting the data so that they limit their interpretation. This is a valid approach, as their curve fitting was inappropriate. The current data presentation is acceptable.

Critique: The authors transition to the use of actinomycin D in the end of the manuscript, at Figure 6. If stress causes p53 stabilization, then this should be observed with H₂O₂ treatment as well. All previous experiments were carried out with peroxide. Actinomycin D is highly toxic, affects the production of all proteins through its inhibition of RNA polymerase, and thus results obtained with actinomycin D can be artifactual. All experiments should at least be carried out with the same compound (H₂O₂).

Response: the authors have added additional data with H₂O₂.

Critique: The authors refer to the use of Actinomycin D in clinical trials. How does the dose used in their experiments compare with the clinically used dose?

Response: The authors answered this question.

D. Appropriate use of statistics and treatment of uncertainties:

Critiques: The authors should clarify the statistical significance designation in Figure 1. For example, they should indicate whether any of the NAC-treated cells are significantly different from control NAC-treated cells. It is noted that in the presence of UV, the treated sample appears to be higher than the control sample. In the last sentence on page 8 the authors state that the modification "apparently impeded the association of the interface with the potential nuclear macromolecules", however the data shown is a simulation and should only be stated to suggest a mechanism.

Response: Authors followed the reviewer's suggestions.

E. Conclusions: robustness, validity, reliability

Critiques: In the first full paragraph of page 6 of the manuscript the authors suggest that the observations in the presence of DTT have a strong causal implication, however these observations are merely correlations. It is suggested that this sentence be reworded. On page 9 the authors imply their observations of cells treated with RNAs and DNAs to show a causal relationship when in fact this is an association under conditions that are highly abnormal. The text should be more carefully worded using the words "suggest" rather than "confirmed". At the top of page 10, beginning of the paragraph, the authors should use the word "test" as opposed to "verify".

Response: Authors followed the reviewer's suggestions.

F. Suggested improvements: experiments, data for possible revision:

Please see all issues mentioned above. In addition:

Critiques: The authors should use the HUGO nomenclature NPM1 for the gene nucleophosmin. The use of the English language by the authors could use some polishing and reviewed by a native speaker is recommended. The authors are encouraged to use different color schemes for different treatments in Figure 5C. The use of similar colors for different treatments is confusing.

Response: Authors followed the reviewer's suggestions.

G. References: appropriate credit to previous work?

The authors appear to reference previous work correctly.

H. Clarity and context:

Critiques: The abstract requires clarification in that it is unclear that the authors are referring to a different mutant (than the Cys275 one) when they discuss "the mutant protein with nucleoplasmic location". In the discussion, the following point should be clearly addressed: How does the dose Actinomycin D dose used in their experiments compare with the clinically used dose?

Response: Authors followed the reviewer's suggestions.

In summary, the authors' responses to the critiques on data and methodology were acceptable. Clearly, the authors put a strong effort into addressing all the reviewer's criticisms. The manuscript has improved considerably.

---We are grateful to this reviewer for his/her acceptance of all efforts and approaches we made in the revision.

Reviewer #5 (Remarks to the Author):

In their revised submission, the Authors addressed several of our concerns; however, we still feel that this work does not add further insights to the already known mechanism of p53 stabilization.

In detail:

1. We had asked the Authors to assess the role of ARF (which has a key function in the p53-mdm2-NPM1 pathway) by using a cell line expressing it, since U2OS are ARF negative.

The Authors state: "During the study we couldn't see the nucleolar localization of ARF in HeLa cells; ARF was intensely stained only in the nucleoplasm". Since this result is in contrast with published reports, did the Authors consider that this observation might have been due to non-specific staining? We are also surprised that the Authors chose HeLa cells where p53 activity levels are known to be difficult to establish due to inhibition by the HPV oncoprotein E6.

--- HeLa cells were observed in early stage of this study. We were puzzled by the results of staining that were inconsistent with the published report. Now we used U2OS cells that were enforcedly expressed with ARF following this reviewer's suggestion, which provided new clues to ARF-NPM1-p53 interrelationship.

The Authors do not specify which cell lines they received from colleagues to try to assess ARF role; were they able to test the U2OS/Tet-On/p19ARF-inducible cell line (doi:10.4161/auto.25831), or the inducible MT-Arf cell line (a NIH-3T3 cell

derivative; doi:10.1101/gad.1213904)?

---We tried to ask for the U2OS/Tet-On/p19ARF-inducible cell line (doi:10.4161/auto.25831), or the inducible MT-Arf cell line (a NIH-3T3 cell derivative; doi:10.1101/gad.1213904) from the authors by emails but unfortunately got no reply. Lucky enough, we were given plasmids of p14/ARF with tag as a gift from a colleague who previously worked on ARF [EMBO J. 2009 Jan 7;28(1):21-33]. The ARF plasmids were transiently expressed in U2OS cells, and transfection efficiency was about 50%. Experiments based on these cells provided some interesting data that addressed this reviewer's concerns.

The results of immunofluorescence using antibodies against HA tag showed that exogenous ARF was distributed both in the nucleolus and nucleoplasm in U2OS cells (as *new Supplementary Fig. 6e*, fibrillarin served as a nucleolar marker). When visualizing endogenous NPM1, we found that the levels of HPM1 in those cells that expressed higher ectopic ARF were markedly reduced, which was consistent with the notion claimed by previous documents that NPM1 is degraded by ARF [Mol Cell. 2003 Nov;12(5):1151-64.]. Since this finding was not new and irrelevant to our major focus, we didn't show the data, and didn't discuss it. But, this made us unable to assess dual staining on the same cells. Alternatively, we examined the localization of the HA-ARF and endogenous NPM1 in the respective cells came from a same coverslip, and we found that the nucleolar ARF did not move out following Act.D treatment, and in contrast, the nucleolus/nucleoplasm ratio of ARF was slightly increased, while NPM1 translocate to the nucleoplasm under the same condition (as *new Fig. 6g*). We determined the levels of p53 by immunoblotting and found that, compared with the mock DNA, overexpression of ARF led to an increased accumulation of p53 under non-stress condition, which was true no matter that U2OS cells were with normal NPM1 or knockdown of NPM1. However, after cells were exposed to Act.D, overexpression of ARF alone did not produce further accumulation of p53 in NPM1-knockdown cells. An enhanced p53 accumulation was observed when the WT NPM1 was added-back, but adding-back of the C275S mutant still could not causes the increase of p53 (as *new Fig. 6h*). These data suggest that the nucleoplasmic fraction of ARF alone is able to induce p53 accumulation under basal conditions; the further accumulation under stress conditions is determined by the nucleoplasmic NPM1. This idea was substantiated by the results of co-IP in the presence of Act.D treatment. We found that the disruption of the HDM2-p53 interaction occurred in the WT NPM1- but not C275S NPM1 adding-back cells, while ARF binding with HDM2 remained same in both cell groups (as *new Fig. 6i*). This means that although ARF expression can promote p53 stabilization, in line with previous studies, it holds true in U2OS cells free of the nucleolar stress; the enhancement of p53 accumulation under stress condition requires NPM1 translocation to the nucleoplasm. Given that there existed complicated reciprocal regulations between ARF and NPM1, it is beyond our capacity to clarify all the issues. However, previous studies in literature seldom applied image approaches to examine ARF distribution following stress. To our

knowledge only one paper [Cancer Res. 2005 Nov 1;65(21):9834-42.] reported the nucleoplasmic translocation ARF in DU145 cells after DNA damage stress, and their model that claimed necessity of ARF translocation in NPM1 action on p53-HDM2 has not been widely discussed or cited in subsequent reviews. In the present study we at least clarify that, unlike NPM1, ARF doesn't increase its nucleoplasmic fraction following a typical nucleolar stress. Moreover, under stress condition, the amounts of ARF bound to HDM2 remain similar in cells bearing the WT or mutant NPM1. Therefore, we concluded that irrespective of ARF, nucleoplasmic translocation of NPM1 plays an indispensable role that connects nucleolar stress to p53 activation.

We appreciate this reviewer's instructive suggestions which make our understanding to the ARF-NPM1-p53 interrelationship increased.

2. *Fig.6/Figure for reviewer 5 a: The indicated experiment only shows p53 (not NPM or NPM mutant) thus preventing any correlation between NPM and p53 localization*
---We had intended to show subcellular compartment where p53 accumulated in images in Fig.6, which especially allowed examining the accumulation of p53 in individual cells corresponding to NPM WT/ mutant expression. Figure for reviewer 5 a was only to exclude the cytoplasmic accumulation of 53.

3. *Fig. S5: We had suggested that the Authors could use as negative control a region of rDNA or rRNA where they did not expect to find NPM1 binding.*

In their rebuttal the Authors state: "We noticed during the ChIP work that NPM1 had a wild but somewhere weaker binding with the full-length rDNA genes; apart from the stronger binding at the specific elements. The data were presented for this reviewer as Figure for reviewer 5 e. This was the reason we didn't choose a rDNA element as the negative control."

However, as the 35S ribosomal DNA (rDNA) units, repeated in tandem at one or more chromosomal loci, are separated by an intergenic spacer (IGS) and the IGS is a repeated but not transcribed region, we believe it could have been used a negative control.

---We have two reasons for using GAPDH region as the negative control. The human ribosomal cassette is composed of a ~13 kb transcribed region followed by a non-transcribed ~30 kb intergenic spacer (IGS) (Figure for reviewer 5 a). For mapping the binding capacity of NPM1 with the whole human ribosomal cassette, we repeated the same experiment as Murano, K. performed [Mol Cell Biol. 2008 May;28(10):3114-26.] using 16 primers of which up to 10 covered the non-transcriptional IGS region (Figure for reviewer 5 b). Similar to their results (Figure for reviewer 5 c), our results, using the similar 16 primers as they designed, showed all of the regions mapped in ChIP assay were associated with B23 (NPM1), although the IGS region had a weaker binding capacity (Figure for reviewer 5 d). That was the reason why the primer specific for IGS sequence was not suitable. Secondly, we accessed the fragments content of GAPDH and rDNA regions after immunoprecipitation (Figure for reviewer 5 e). The results showed

that the content of GAPDH fragment are comparable in Mock and FLAG-NPM1 groups; whereas the primer I fragment was accumulated to ~4 folds in FLAG-NPM1 group. Thus, we used GAPDH as a negative control region that NPM1 does not bind.

Figures for reviewer 5 | (a) The schematic for human rDNA sequences, composed of a ~13 kb transcribed region followed by a non-transcribed ~30 kb intergenic spacer, cited from ⁽¹⁾. **(b)** Primers used in ChIP assay, primers I to VI covered transcribed region (0 ~ 13,000 bp), and other primers covered IGS region (13,000 ~ 43,000 bp), as same as designed from ⁽²⁾. **(c)** Results cited from ⁽²⁾. **(d)** Folds of rDNA bound to NPM1. **(e)** The contents of GAPDH and primer I fragments immunoprecipitated with Mock and FLAG-NPM1.

References

- Jacob MD, Audas TE, Mullineux ST, Lee S. Where no RNA polymerase has gone before: novel functional transcripts derived from the ribosomal intergenic spacer. *Nucleus* 3, 315-319 (2012).
- Murano K, Okuwaki M, Hisaoka M, Nagata K. Transcription regulation of the rRNA gene by a multifunctional nucleolar protein, B23/nucleophosmin, through its histone chaperone activity. *Molecular and cellular biology* 28, 3114-3126 (2008).

REVIEWERS' COMMENTS:

Reviewer #1 (Remarks to the Author):

I find that the authors have addressed my concerns.

Reviewer #2 only had comments to the editor.

Reviewer #5 (Remarks to the Author):

The Authors have now provided additional experiments/figures and adequately addressed our concerns/suggestions.

Response to Referees

REVIEWERS' COMMENTS:

Reviewer #1 (Remarks to the Author):

I find that the authors have addressed my concerns.

Reviewer #2 only had comments to the editor.

Reviewer #5 (Remarks to the Author):

The Authors have now provided additional experiments/figures and adequately addressed our concerns/suggestions.

--We appreciate that all reviewers accepted our revisions and explanations.